# Surrogate Modeling of 3D Rayleigh-Bénard Convection with Equivariant Autoencoders

## Abstract

The use of machine learning for modeling, understanding, and controlling large-scale physics systems is quickly gaining in popularity, with examples ranging from electromagnetism over nuclear fusion reactors and magneto-hydrodynamics to fluid mechanics and climate modeling. These systems—governed by partial differential equations—present unique challenges regarding the large number of degrees of freedom and the complex dynamics over many scales both in space and time, and additional measures to improve accuracy and sample efficiency are highly desirable. We present an end-to-end equivariant surrogate model consisting of an equivariant convolutional autoencoder and an equivariant convolutional LSTM using $G$-steerable kernels. As a case study, we consider the three-dimensional Rayleigh-Bénard convection, which describes the buoyancy-driven fluid flow between a heated bottom and a cooled top plate. While the system is E(2)-equivariant in the horizontal plane, the boundary conditions break the translational equivariance in the vertical direction. Our architecture leverages vertically stacked layers of $D_4$-steerable kernels, with additional partial kernel sharing in the vertical direction for further efficiency improvement. We demonstrate significant gains in sample and parameter efficiency, as well as a better scaling to more complex dynamics.

## 1 Introduction

The ability to perform fast and efficient simulations of large-scale physics systems governed by partial differential equations (PDEs) is of vital importance in many areas of science and engineering. As in almost any other area, machine learning plays an increasingly important role, where scenarios of great interest are real-time prediction, uncertainty quantification, optimization, and control. Application areas include weather forecasting (Kurth et al., 2023) and climate modeling (Vlachas et al., 2018), aerodynamics and fluid mechanics (Brunton et al., 2020), combustion (Ihme et al., 2022), or the plasma in nuclear fusion reactors (Kates-Harbeck et al., 2019). PDE-governed systems often exhibit complex or chaotic behavior over a vast range of scales in both space and time. Along with the very large number of degrees of freedom (after discretization using, e.g., finite elements), this renders the resulting time series particularly challenging for surrogate modeling, especially in multi-query contexts such as prediction and control (Bieker et al., 2020). Fortunately, many dynamical systems evolve on a low-dimensional manifold (e.g., an attractor), which allows for dimensionality reduction techniques and surrogate modeling. As linear approximation techniques such as proper orthogonal decomposition (POD) (Sirovich, 1987) tend to break down once the dynamics become more complex—as is the case for turbulent flows—nonlinear variants such as autoencoders have recently become more and more important in the physics modeling community (Nikolopoulos et al., 2022; Francés-Belda et al., 2024). However, the resulting reduced spaces are less interpretable, and the training is much more data-hungry and sensitive to hyperparameter tuning. The goal of this paper is to include known symmetries in the surrogate modeling process of complicated 3D physics simulations via autoencoders. Studying a prototypical convection or climate system described as Rayleigh-Bénard convection, our contributions are the following (cf. Fig. 1 for a sketch):

- Whereas most papers are concerned with 2D in space, we develop an end-to-end equivariant architecture for the much more challenging prediction of 3D time-dependent PDEs, consisting of an autoencoder and an LSTM for time series prediction in latent space. To preserve the symmetry, we omit latent space flattening but preserve the original 3D structure of the problem.

Figure 1: **Architecture overview.** An initial snapshot $s_t$ is encoded via our E(2) equivariant 3D autoencoder to the latent representation $z_t$; $z_t$ is evolved forward in time to $\tilde{z}_{t+1}$ via our equivariant LSTM; $\tilde{z}_{t+1}$ is decoded via our E(2) equivariant 3D decoder to yield a predicted next snapshot $\tilde{s}_{t+1}$.

- The system is equivariant under rotations, reflections, and translations (i.e., the symmetry group $E(n)$), but only in the horizontal plane due to the buoyancy. We introduce an efficient $G$-steerable autoencoder architecture that respects the symmetry group $\mathbb{Z}^2 \rtimes D_4$ in the horizontal plane (the subgroup of $E(2)$ with shifts on a grid, reflections, and discrete 90-degree rotations).
- Additional local kernel sharing in the vertical direction further improves the parameter efficiency.
- We demonstrate that for large-scale systems, a separate training of encoding and time stepping can be computationally much more efficient than end-to-end training.
- We demonstrate high accuracy at a compression rate $> 98\%$, while saving one order of magnitude in both trainable parameters and training data compared to non-symmetric architectures.

## 2 RELATED WORK

**Rayleigh-Bénard convection** (Pandey et al., 2018) models the dynamics of a compressible fluid between two flat plates, where the bottom plate is heated while the top plate is cooled. This induces buoyancy forces, which in turn result in fluid motion. At moderate Rayleigh numbers $Ra$ (a dimensionless parameter quantifying the driving force induced by the temperature difference), one observes well-characterized convection rolls. With increasing $Ra$, the fluid becomes turbulent which results in a large number of vortices on varying time and spatial scales, rendering the fluid hard to predict and characterize (Vieweg et al., 2021).

In recent years, a large number of works have appeared on **surrogate modeling** of PDE systems. Many approaches rely on the identification of a low-dimensional latent space, for instance, via POD (Soucasse et al., 2019) or *autoencoders* (Pandey et al., 2022; Akbari et al., 2022; de Sousa Almeida et al., 2023). Alternatively, the direct prediction of the full state can be accelerated using linear methods such as the *Koopman operator* framework (Klus et al., 2020; Markmann et al., 2024; Azencot et al., 2020; Nayak et al., 2025) or physics-informed machine learning (Karniadakis et al., 2021; Clark Di Leoni et al., 2023; Hammoud et al., 2023). In that latter category, *neural operators* (Li et al., 2021; Lu et al., 2021; Kovachki et al., 2023; Goswami et al., 2023; Straat et al., 2025) are very prominent, and have been demonstrated to show great performance on a large number of systems. Finally, there have been great advances in the deep learning area as well, most prominently using *U-Nets* (Gupta & Brandstetter, 2023; Lei & Li, 2025), *transformer* architectures (Gao et al., 2024; Holzschuh et al., 2025b), and generative frameworks such as *GANs* (Chen et al., 2020) or *diffusion models* (Holzschuh et al., 2025a; Bastek et al., 2025; Li et al., 2025; Oommen et al., 2025).

The **exploitation of symmetries** has recently become increasingly popular, and it is now often referred to under the umbrella term geometric deep learning (Bronstein et al., 2021). Most of the literature in this area is until now related to classical learning tasks such as image classification (Cohen & Welling, 2016; Esteves et al., 2018a;b; Weiler & Cesa, 2019; Bronstein et al., 2021; Weiler et al., 2025). However, equivariant architectures have been developed for various other tasks, such as the analysis of graph-structured data (e.g., molecules (Wu et al., 2021)), or for the prediction of PDEs (Jenner & Weiler, 2022; Zhdanov et al., 2024; Harder et al., 2024b). Equivariant autoencoder architectures for dimensionality reduction were proposed in, e.g., Kuzminykh et al. (2018); Guo et al. (2019); Huang et al. (2022), see also Hao et al. (2023); Yasuda & Onishi (2023) for applications to fluid flows. Further examples of equivariant learning of PDEs were presented in various contexts, such as Koopman operator theory and Dynamic Mode Decomposition (Salova et al., 2019; Baddoo et al., 2023; Harder et al., 2024a; Peitz et al., 2025), as well as reinforcement learning (Vignon et al., 2023; Vasanth et al., 2024; Peitz et al., 2024; Jeon et al., 2024).

## 3 PRELIMINARIES

Partial differential equations (PDEs) describe dynamical systems whose state $s$ is a function of multiple variables such as space $x \in \Omega \subset \mathbb{R}^n$ and time $t \in \mathbb{R}^{\geq 0}$. The equations of motion are described by (nonlinear) partial differential operators,

$$\frac{\partial s}{\partial t} = \mathcal{F}(s, \nabla s, \Delta s, \dots) \qquad \text{for } x \in \Omega, \ t \in \mathbb{R}^{\geq 0},$$

accompanied by appropriate boundary conditions on $\Gamma = \partial \Omega$ and initial conditions. In many cases, $s$ is *equivariant* with respect to certain symmetry transformations such as translations or rotations.

### 3.1 SYMMETRIES

We here give a very brief overview of symmetry groups and group actions; more detailed introductions can be found in, e.g., Weiler et al. (2021); Bronstein et al. (2021). A group is a tuple $(G, \circ)$, where $G$ is a set and $\circ : G \times G \to G, (g, h) \mapsto gh$ an operation which is associative, has an identity element $e$ and inverses (denoted by $g^{-1}$ for $g \in G$). The group operation describes the effect of chaining symmetry transformations. However, to employ group theory in practice, one needs an additional *object* that the group can *act* on. A *group action* is a function $G \times X \to X, (g, x) \mapsto g \cdot x$, where $X$ is the underlying set of objects that are transformed. As for the group operation, one assumes associativity in the sense that $g \cdot (h \cdot x) = (gh) \cdot x$ together with invariance under the identity element.

A linear representation of $G$ on a vector space $V$ is a tuple $(\rho, V)$, where $\rho : G \to GL(V)$ is a group homomorphism from $G$ to the general linear group $GL(V)$ of invertible linear maps of the vector space $V$, see (Weiler et al., 2021, Appendix B.5) for details. In case $V = \mathbb{R}^n$, the group action is defined as matrix multiplication by $\rho(g)$, i.e., $\rho(g) = A \in \mathbb{R}^{n \times n}, (g, x) \mapsto Ax$.

If a group action is defined on $X$, one can obtain an action on the space of functions of the form $\phi : X \to \mathbb{R}^m$ by introducing $(\rho(g)\phi)(x) := \phi(g^{-1} \cdot x)$. Here, we have already denoted the action as a representation, see (Cohen & Welling, 2017), as it is linear.

**Example 1.** *The 2D Euclidean group $E(2)$ is the group of planar translations, rotations, and reflections. It can be expressed as the semidirect product of translations $(\mathbb{R}^2, +)$ and orthogonal transformations, i.e., $E(2) = (\mathbb{R}^2, +) \rtimes O(2)$. A linear representation of $E(2)$ can be expressed as*

$$E(2) = \left\{ \begin{pmatrix} A & \tau \\ 0 & 1 \end{pmatrix} \middle| A \in O(2), \tau \in \mathbb{R}^2 \right\}, \quad \text{with } O(2) = \left\{ A \in \mathbb{R}^{2 \times 2} \middle| A^\top A = Id \right\}.$$

*An element $h \in E(2)$ can be decomposed into $h = \tau g$, where $\tau$ is a pure translation and $g$ is a transformation that leaves the origin invariant.*

In a straightforward manner, we can consider *subgroups* $H \leq E(2)$ when replacing $O(2)$ by a subgroup $G \leq O(2)$ and defining $H = (\mathbb{R}^2, +) \rtimes G$. When working with data on structured grids, as we do here, discrete translations and rotations can be implemented in a particularly efficient manner. This results in the dihedral group $D_4$, which allows for flips and 90-degree rotations. In combination with quantized translations on the grid nodes, we obtain $H = (\mathbb{Z}^2, +) \rtimes D_4 < E(2)$.

In general, for two sets $X$ and $Y$, a function $\phi : X \to Y$ is called equivariant if there is a group acting both on $X$ and $Y$ such that $\phi(g \cdot x) = g \cdot \phi(x)$, or, equivalently, $\phi(x) = g^{-1} \cdot \phi(g \cdot x)$, for all $x \in X$ and $g \in G$. Intuitively, this means that one can obtain the result of $\phi(x)$ by first evaluating $\phi$ at the transformed object $g \cdot x$ and then applying the inverse transformation $g^{-1}$ afterwards, cf. Fig. 2 for an illustration using the Rayleigh-Bénard system described in Section 3.2.

**Convolutions.** Conventional convolutions convolve a feature map $f_{\text{in}} : \mathbb{R}^n \to \mathbb{R}^{C_{\text{in}}}$ with a kernel $\psi : \mathbb{R}^n \to \mathbb{R}^{C_{\text{out}} \times C_{\text{in}}}$ as follows in order to produce a feature map $f_{\text{out}} : \mathbb{R}^n \to \mathbb{R}^{C_{\text{out}}}$:

$$f_{\text{out}}(x) = [f_{\text{in}} * \psi](x) = \int_{\mathbb{R}^n} \psi(x - y) f_{\text{in}}(y) \, dy$$

This is an example of a translation equivariant operation, as $[\rho(\tau)f_{\text{in}}] * \psi = \rho(\tau)[f_{\text{in}} * \psi]$ for $\tau \in \mathbb{R}^n$ (Cohen & Welling, 2016).

**Steerable convolutions.** In steerable CNNs, we define feature spaces in such a manner that the respective feature fields $f : \mathbb{R}^n \to \mathbb{R}^C$ are *steerable* (Cohen & Welling, 2017; Weiler & Cesa, 2019). This means that for each $n$-dimensional input $x \in \mathbb{R}^n$, each $C$-dimensional feature $f(x) \in \mathbb{R}^C$ transforms under the group action of a given group $G$ (e.g., $O(n)$)—the translational equivariance is automatically satisfied when the parameters of the learnable kernel $\psi$ are position-independent (Weiler et al., 2021). In particular, this ensures that the transformation of vector-valued features transforms their orientation according to the group action. As a consequence, all CNN architectures that are

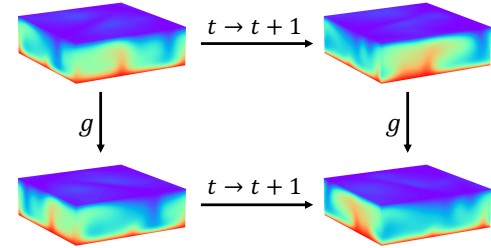

Figure 2: **Equivariance of time evolution and rotation.** The equivariance of the time evolution of the temperature field under a 90-degree rotation $g$ is illustrated by the commutativity diagram.

constructed using $G$-steerable kernels are equivariant with respect to the group $H = (\mathbb{R}^n, +) \rtimes G$. In Weiler & Cesa (2019), it was shown that for this property to hold, a kernel $\psi : \mathbb{R}^2 \to \mathbb{R}^{C_{\text{out}} \times C_{\text{in}}}$ has to satisfy the steerability constraint

$$\psi(g \cdot x) = \rho_{\text{out}}(g)\psi(x)\rho_{\text{in}}(g^{-1}) \quad \forall g \in G, x \in \mathbb{R}^2. \tag{1}$$

For a network to be equivariant, the kernel constraint (1) has to hold for all combinations of $\rho_{\text{in}}$ and $\rho_{\text{out}}$. Instead, it is shown in Weiler & Cesa (2019) that a much simpler approach is to introduce a change of basis $Q$, by which any $\rho$ can be decomposed into the direct sum of its irreducible representations (irreps), i.e., $\rho = Q^{-1} \left[ \oplus_{i \in I} \psi_i \right] Q$. One can thus replace (1) by individual constraints on the irreps. Since we have $G \leq O(2)$, $G$ is norm preserving, such that the kernel constraint can further be reformulated in terms of a Fourier series expansion of the kernel, ultimately resulting in a set of constraints on the Fourier coefficients, cf. Weiler & Cesa (2019); Weiler et al. (2021) for detailed derivations and discussions. We will make heavy use of this approach in our architecture, both in terms of the autoencoder and the LSTM for time series prediction.

## 3.2 RAYLEIGH-BÉNARD CONVECTION

Rayleigh-Bénard convection describes the flow between two flat plates. The bottom plate is heated, while the top plate is cooled, which induces buoyancy forces that cause the fluid to move in convection rolls, cf. Figs. 1 or 2 for illustrations. Depending on the driving force—the temperature difference, encoded by the dimensionless Rayleigh number $Ra$—the system is deterministic at first, then becomes increasingly complex and turbulent for larger $Ra$ (Pandey et al., 2018). More details about the specific PDE and the numerical simulations can be found in Appendix A.1. A more detailed discussion on the system's symmetries—including a proof—can be found in Appendix A.2. Moreover, a list of other systems with broken symmetries can be found in Appendix A.3.

In the following, we will summarize the quantities of interest—temperature $T(x, t)$ and velocity $u(x, t)$, but not the pressure—in the state vector $s(x, t) \in \mathbb{R}^{C_{\text{in}}}$, where $C_{\text{in}} = 4$ is the number of input channels. Due to the discretization in space, the state function becomes a large tensor of dimension $N = N_1 \times N_2 \times N_3$. Moreover, we will consider snapshots at discrete times $t \in \mathbb{N}$, that is, $s_t = (u_t, T_t) \in \mathbb{R}^{N_1 \times N_2 \times N_3 \times C_{\text{in}}}$.

## 4 METHODS

Our framework comprises two main components: a convolutional autoencoder and a convolutional LSTM, trained independently. As illustrated in Fig. 1, the autoencoder first encodes a snapshot $s_t$ into a latent representation $z_t$. The LSTM then sequentially forecasts subsequent latent representations $\tilde{z}_{t+1}, \tilde{z}_{t+2}, \ldots$, which are subsequently decoded to full-state snapshots, $\tilde{s}_{t+1}, \tilde{s}_{t+2}, \ldots$. Both the autoencoder and LSTM are designed in an equivariant fashion. As a consequence, the entire framework is end-to-end equivariant.

### 4.1 3D STEERABLE CONVOLUTION ON E(2)

As discussed in Appendix A.2, the 3D Rayleigh-Bénard convection is $E(2)$-equivariant in the horizontal plane, which we enforce by steerable convolutions, while introducing height-dependent kernels that adapt to the varying dynamics at different heights.

#### 4.1.1 STEERABLE CONVOLUTION

The system state $s$ consists of both scalar temperature and vector-valued velocity fields. Under transformation of these fields by $\tau g \in E(2)$, the temperature field $T : \mathbb{R}^3 \to \mathbb{R}$ transforms as $T(x) \mapsto 1 \cdot T(g^{-1}(x - \tau))$, whereas the velocity-field $u : \mathbb{R}^3 \to \mathbb{R}^3$ transforms as $u(x) \mapsto g \cdot u(g^{-1}(x - \tau))$. Note that the velocity vectors are themselves transformed via $g$ to preserve their orientation as the field is transformed (Cohen & Welling, 2017; Weiler & Cesa, 2019).

To ensure equivariant mappings between three-dimensional feature fields $f_{\text{in}} : \mathbb{R}^3 \to \mathbb{R}^{C_{\text{in}}}$ and $f_{\text{out}} : \mathbb{R}^3 \to \mathbb{R}^{C_{\text{out}}}$, with corresponding transformations $\rho_{\text{in}}$ and $\rho_{\text{out}}$, we constrain the kernels $\psi : \mathbb{R}^3 \to \mathbb{R}^{C_{\text{out}} \times C_{\text{in}}}$ to be $O(2)$-steerable. Since the equivariance is restricted to the horizontal plane (i.e., $x_1$ and $x_2$) the group $O(2)$ acts on points $x$ via the block-diagonal representation $\rho(g) = A \oplus 1$. The constraint can thus be decomposed into independent constraints for every fixed height $\hat{x}_3$:

$$\psi\left(g \cdot \begin{pmatrix} x_1 \\ x_2 \\ \hat{x}_3 \end{pmatrix}\right) = \rho_{\text{out}}(g)\psi\left(\begin{pmatrix} x_1 \\ x_2 \\ \hat{x}_3 \end{pmatrix}\right)\rho_{\text{in}}(g^{-1}) \quad \forall g \in O(2), \begin{pmatrix} x_1 \\ x_2 \end{pmatrix} \in \mathbb{R}^2. \tag{2}$$

Thus, a three-dimensional $O(2)$-steerable kernel can be constructed as a stack of 2D $O(2)$-steerable kernels. For these, Weiler & Cesa (2019) have solved the steerability constraint as well as for important subgroups such as $C_4$ and $D_4$. This result has been applied in our implementation to efficiently design height-dependent kernels utilizing the PyTorch-based library escnn.[1] For computational reasons, we restrict our implementation to the subgroup $H = (\mathbb{Z}^2, +) \rtimes D_4 < E(2)$, as this optimally corresponds to our data on a rectangular grid. Thus, we will from now on consider discretized kernels and feature fields on $\mathbb{Z}^3$. The case of continuous rotations according to $O(2)$ would require interpolation between grid points (see, e.g., Esteves et al. (2018a)). In our experiments, this resulted in inferior performance compared to the grid-consistent 90-degree rotations and flips.

Within our framework, we use various types of feature fields. Both the input to the AE-encoder and the output of the decoder are composed of the scalar field $T$ and the vector field $u$. All intermediate representations, however, make use of regular feature fields, which transform under the regular representation by permuting the channels. Steerable convolutions between regular feature fields are equivalent to regular group convolutions (Cohen & Welling, 2016), which apply the same kernel in every orientation $g \in G$, resulting in the $|G|$-dimensional regular feature fields.

#### 4.1.2 3D CONVOLUTIONS WITH HEIGHT-DEPENDENT KERNELS

Height-dependent features—such as temperature or velocity patterns—play a critical role in modeling the system. As a result, applying the same kernel across all heights—as would be the case for a regular 3D CNN—is insufficient to capture the system's vertical dynamics. To address this limitation, we modify the conventional 3D convolutions by learning height-dependent steerable kernels. While this ensures horizontal parameter sharing, we allow the convolution operation to adapt to the distinct features at each height, ensuring that the vertical structure is captured effectively. For computing the output feature map's value at position $x = (x_1, x_2, x_3) \in \mathbb{Z}^3$, the input $f : \mathbb{Z}^3 \to \mathbb{R}^{C_{\text{in}}}$ is convolved with the height-dependent kernel $\psi_{x_3} : \mathbb{Z}^3 \to \mathbb{R}^{C_{\text{out}} \times C_{\text{in}}}$ via $[f * \psi](x) = \sum_{y \in \mathbb{Z}^3} \psi_{x_3}(x - y)f(y)$.

**Local vertical parameter sharing.** Although the Rayleigh-Bénard system does not exhibit global vertical translation equivariance, it approximately maintains this property within a local neighborhood (see Appendix A.2). This suggests that features at a given height $x_3$ are locally correlated with features at vertical positions within a $2k + 1$-sized neighborhood $\mathcal{N}(x_3) = \{x_3 - k, \ldots, x_3 + k\}$. This approximate local equivariance in the vertical direction can be exploited by choosing a smaller number of channels in each layer, but then applying these learned kernels across the local neighborhood $\mathcal{N}$, thereby again increasing the number of output channels, cf. Fig. 3 for a sketch. This parameter reduction leads to a more efficient and scalable model, rendering it suitable for simulating larger-scale

---

[1]https://github.com/QUVA-Lab/escnn

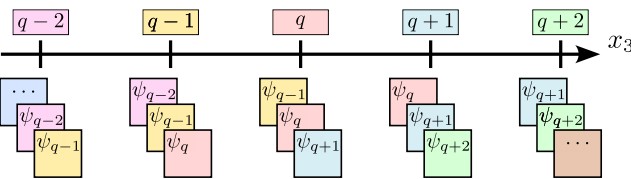

Figure 3: **Local vertical kernel sharing.** Kernels of different heights $\hat{x}_3 = q - 2, q - 1, q, \ldots$ are applied to the neighboring $k = 1$ heights, resulting in a total of 3 kernels being applied each.

systems. As described in Appendix B.2, 3D convolutions with height-dependent kernels and local vertical parameter sharing can be efficiently implemented by wrapping 2D (steerable) convolutions.

### 4.2 EQUIVARIANT AUTOENCODER

The convolutional autoencoder (CAE) is designed to respect the horizontal $E(2)$ symmetry by incorporating equivariant convolutions into both the encoder and decoder. In the encoder, an input snapshot $s_t$ of shape $N_1 \times N_2 \times N_3 \times C_{\mathsf{in}}$ is mapped to a lower-dimensional latent representation $z_t$ of shape $M_1 \times M_2 \times M_3 \times C_{\mathsf{latent}}$, through a sequence of convolutional layers progressively extracting higher-level features. Each layer is followed by activation functions and max-pooling. Notably—instead of the common flattening—the latent space maintains the systems original spatial structure to preserve spatial correlations in the data, although we have $M \ll N$. The number of channels $C_{\mathsf{latent}}$ is used to control the size and expressiveness of the latent representation.

The decoder applies a series of convolutions, followed by activation functions. Pooling operations are replaced by upsampling. To preserve continuity while upsampling, we employ trilinear interpolation. Throughout the entire framework (CAE + LSTM), padding is applied to the convolutional layers to ensure that the spatial dimensions of the data are only altered by pooling and upsampling operations. For the horizontal dimensions, circular padding is used to match the periodic boundary conditions of the Rayleigh-Bénard system, while vertical dimensions use zero padding. Note that without padding, the decoder would require transposed convolutions (Dumoulin & Visin, 2018) to reverse the dimensional changes induced by convolutions without padding.

### 4.3 EQUIVARIANT LSTM

Long short-term memory (LSTM) networks (Hochreiter & Schmidhuber, 1997) extend conventional recurrent neural networks (RNNs) for processing sequential data by an additional cell state $c_t$. While the hidden state $h_t$ stores currently relevant information for predicting the next time step, the cell state $c_t$ captures long-term information over long sequences. The content of $c_t$ is regulated by the input gate $i_t$, which controls which information is added, and the forget gate $f_t$, which controls how much information is discarded. The output gate $o_t$ determines which information is passed from $c_t$ to $h_t$. To preserve the spatial structure of the latent space $z$, we use convolutional LSTMs (Shi et al., 2015) that replace the fully connected layers of standard LSTMs with, in our case, equivariant convolutions and also introduce equivariant convolutions in peephole connections, see B.1 for details as well as a proof that the architecture is equivariant.

## 5 EXPERIMENTS

We evaluate the performance of our end-to-end equivariant framework against a baseline model using standard, non-steerable 3D convolutions, with a particular focus on the autoencoder. In addition, we assess the long-term forecasting capabilities of our approach in comparison to 3D Fourier Neural Operators (FNOs) (Li et al., 2021) and 3D U-Nets (Ronneberger et al., 2015; Çiçek et al., 2016) with the same number of parameters, cf. Appendix B.3 for details.

**Datasets.** We generated a dataset of 100 randomly initialized 3D Rayleigh-Bénard convection simulations with $Ra = 2500$ and $Pr = 0.7$, standardized to zero mean and unit standard deviation. The dataset was split into 60 training, 20 validation, and 20 test simulations, each with 400 snapshots in the time interval $t \in [100, 300]$ at a step size of 0.5 and a spatial resolution of $N = 48 \times 48 \times 32$.

Similar datasets were also created for different values of $Ra$ to analyze the effects of data complexity. For long-term forecasting evaluation, we also created an additional dataset of 20 simulations, each containing 1800 snapshots over $t \in [100, 1000]$.

**Architecture.** The CAE encoder and decoder each consist of six convolutional layers with a kernel size of three for the last encoder and first decoder layer, and five for the other layers, applying ELU nonlinearities pointwise.[2] Pooling and upsampling are applied after the encoder and decoder's first, third, and fifth layers, respectively. The latent space has dimensions $M = 6 \times 6 \times 4$ and $C_{\text{latent}} = 32$ channels, resulting in a compression to 1.56% of the original size. Channels roughly double after each pooling operation in the encoder, with the decoder reversing this. Over our experiments, we vary the channel sizes. For a fair comparison, our main models (both standard and equivariant) have approximately the same number of 3.6M parameters. The decoder-only LSTM with one layer of convolutional LSTM cells and an additional convolutional layer for output prediction uses a kernel size of 3. The number of channels is chosen such that the LSTM has approximately 3.7M parameters.

**Training.** CAE training is performed in the standard self-supervised manner on a set of snapshots, where the desired decoder output is the original input. The LSTM was then trained on the latent representations to predict the 50 latent states following the provided sequence of 25 states, with the loss being computed on the latent space. This two-step training approach effectively separates encoding and forecasting, avoiding significant performance drops and speeding up forecasting training by a factor of 20. See Appendix C.1 for training details, and Appendix C.2 for an in-depth discussion of the two-step training approach and the limitations of end-to-end training.

## 5.1 RESULTS

We begin with a detailed evaluation of the CAE performance, followed by an analysis of the long-term forecasting capability of our end-to-end equivariant model.

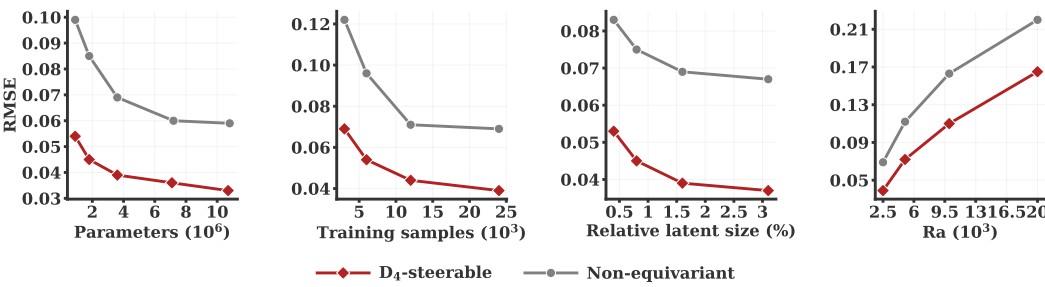

Figure 4: **CAE performances.** Results are shown with varying numbers of parameters, amounts of training samples, compression ratios, and Rayleigh numbers $Ra$. Each plot shows a variation over one of the parameters while keeping the others fixed. The main model has 3.6 million parameters (3rd point), 24,000 training samples (4th point), 1.56% latent size (3rd point) and $Ra = 2500$ (1st point).

**Autoencoder.** Fig. 4 gives an overview of our experiments to compare the equivariant and non-equivariant autoencoders with respect to parameter efficiency, data efficiency, compression capabilities, and scalability to more complex dynamics, i.e., larger $Ra$. Equivariance leads to a significant improvement in reconstruction accuracy, with the RMSE decreasing by 42%, from $0.069 \pm 0.00072$ to $0.04 \pm 0.00093$, where the mean and standard deviation were computed across three models with randomly initialized weights. A comparison to $C_4$-equivariant convolutions shows an RMSE of 0.046 $\pm 0.00164$, showing that both rotations and reflections are relevant for the improved performance.

When studying the four subfigures separately, we observe several significant advantages of incorporating equivariance:

  **(i) Parameter efficiency:** The equivariant model with only 900,000 parameters outperformed the non-equivariant model with 10.8 million parameters, meaning that we obtain an improvement by more than one order of magnitude.

---

[2]All hidden layers in our model transform under the regular representation, which preserves equivariance since it commutes with pointwise nonlinearities (Weiler & Cesa, 2019).

(ii) **Sample efficiency:** The $D_4$ model achieved the same performance as the non-equivariant model when trained on just one-eighth of the training samples, representing a significant reduction in the amount of data required for effective learning, making it particularly valuable in data-limited scenarios or when simulations/experiments are expensive.

(iii) **Compression capabilities:** Even when increasing the compression ratio by a factor of eight, the equivariant model has significantly superior accuracy. This is particularly important when dealing with large-scale systems, as well as for consecutive learning tasks such as training the LSTM.

(iv) **Scaling to complex dynamics:** When increasing the Rayleigh number $Ra$ from $2,500$ up to $20,000$ and retraining the model, we observe that the gap in accuracy even increases further, which indicates that the equivariant model is better equipped to handle more complex dynamics. This suggests that incorporating equivariance into the architecture allows the model to better capture and represent complex fluid flow dynamics inherent in the Rayleigh-Bénard convection system, in particular as the patterns grow more complex.

The results discussed so far have been obtained without local parameter sharing in the vertical direction. Our experiments with local vertical sharing show only minor improvements (cf. Appendix D.7). However, we have considered a fairly moderate number of $N_3 = 32$ vertical layers for now. The results thus merely show that vertical parameter sharing is viable, and we believe that it will become significantly more relevant when considering setups with even larger state spaces. In these situations, learning entirely separate features for each height will become computationally infeasible. A detailed assessment for much larger state spaces will be the focus of future work.

**Long-term forecasting.** We next focus on the evaluation of the long-term forecasting capabilities of our end-to-end equivariant architecture. The model is provided with an input sequence of 50 snapshots and then predicts the subsequent 500 future states in an autoregressive manner, i.e., using its own predictions as input for the next time step.

Fig. 5 shows the median RMSE over time for both our equivariant and non-equivariant models, which was averaged over three separately trained models. The equivariant model consistently outperforms the non-equivariant model by a near-constant margin of approximately 0.05 RMSE across the entire forecast horizon. This indicates that most of the performance gains stem from improved latent representation learned by the equivariant autoencoder, which provides a more stable and informative basis for prediction.

Our method consistently outperforms both FNO and U-Net baselines across all horizons. With equal training horizons (5 steps), it already matches or exceeds their short-term accuracy. The decisive advantage, however, lies in the ability to efficiently train with much longer horizons: while extending FNOs and U-Nets beyond 5 autoregressive steps during training be-

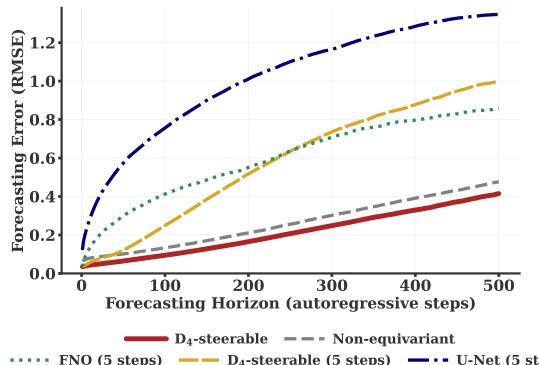

Figure 5: **Forecasting RMSE** averaged across three runs at $Ra = 2500$. FNO and U-Net baselines were restricted to 5 steps due to computational cost, while our model scales efficiently to 50 steps. For reference, we also include the $D_4$-steerable model trained with 5 steps.

comes prohibitively expensive (see Table 1 in Appendix C.1), our latent-space approach scales to 50 steps without difficulty, since forecasting is performed directly in the lower dimensional latent space. This leads to substantially improved long-term forecasts. For completeness, we also report the performance of our model trained with only 5 steps, which still surpasses both baselines at short horizons, although the FNO slightly outperforms our approach at longer horizons.

These results highlight a central advantage of our two-step training strategy: it enables efficient training with long autoregressive horizons in latent space, achieving both superior long-term performance and significantly lower computational cost compared to those baselines.

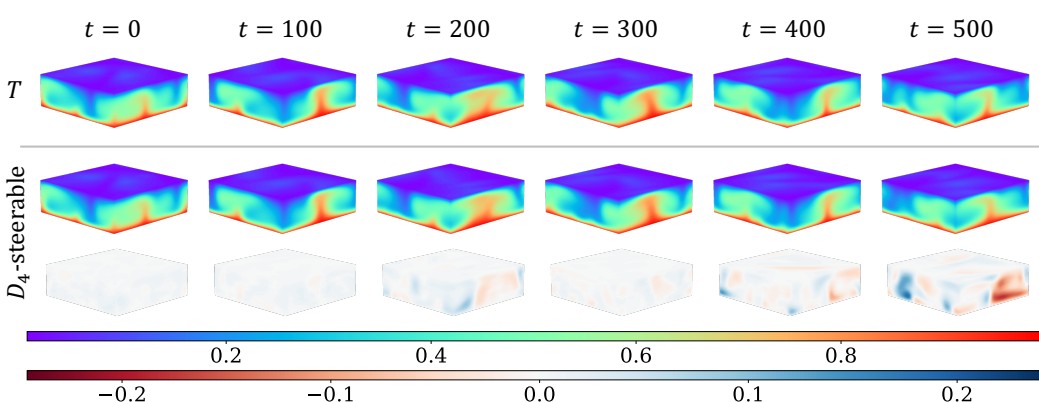

Figure 6: **Representative autoregressive forecast.** The top row displays the ground truth temperature field $T$ at selected time steps $t$. The middle row shows the predictions by the equivariant surrogate model (at $t = 0$, we show CAE reconstruction (encoding + decoding) of the ground truth). The bottom row represents the difference between the predicted and ground truth temperature fields.

Qualitative examples in Fig. 6 further highlight the performance of our surrogate model. The equivariant model is able to preserve the fine-scale structures and large-scale convective patterns over extended time periods. Only minor spatial displacements of the convective plumes are observed, even after hundreds of time steps.

A comprehensive set of supplementary experiments can be found in Appendix D. Finally, we note that the advantages of our surrogate extend beyond comparisons with neural baselines. Appendix C.3 provides a detailed discussion of its benefits over classical PDE solvers.

# 6 CONCLUSION

Our equivariant CAE plus LSTM architecture for efficient surrogate modeling of 3D Rayleigh-Bénard convection consists of horizontally $E(2)$-equivariant kernels that are vertically stacked. Additional local sharing in the vertical direction allows us to increase the number of channels without requiring additional parameters, which further adds to the computational efficiency. We have demonstrated significant advantages in terms of the accuracy and both data and parameter efficiency.

## 6.1 LIMITATIONS AND OUTLOOK

Some points we have not yet discussed or addressed, but believe to be promising for future research:

- We have used idealized simulations without noise or other symmetry-breaking disturbances; this will be essential for studying the robustness in real-world settings.
- In principle, end-to-end training could be superior, as the latent representation is tailored to the dynamics. Since this proved to be challenging and computationally expensive for LSTMs, we believe that a consecutive finetuning phase is more promising. Instead, one could consider linear latent dynamics based on the Koopman operator (Harder et al., 2024a; Azencot et al., 2020).
- In the control setting, equivariant models are very helpful to improve the efficiency, for instance, of world models in the context of reinforcement learning; surrogate modeling for RL of PDEs was studied in, e.g., Werner & Peitz (2024), but a combination with equivariant RL as proposed in van der Pol et al. (2020) has not been investigated.
- For climate applications, equivariant models on spheres would be very interesting to study (cf., e.g., Gastine et al. (2015) for a spherical Rayleigh-Bénard case); besides additional challenges in terms of the numerical implementation, the rotation of the earth would also result in fewer symmetries, such that the usefulness of equivariant models has yet to be determined.
- Incorporating kernels that explicitly depend on control parameters (e.g., Rayleigh number) may improve adaptability and generalization across different flow regimes, including those with qualitative changes such as bifurcations.

REPRODUCIBILITY STATEMENT

We have taken several measures to ensure the reproducibility of our results. A public repository containing the full implementation of our architecture, including training and evaluation scripts, data generation and preprocessing code, as well as detailed instructions for reproducing all experiments, has been released.[3] A complete list of hyperparameter values for each evaluated model will be provided in the repository. The theoretical foundations of our approach are rigorously documented: the symmetries of the 3D Rayleigh–Bénard system are formally proven in Appendix A.2, while the equivariance of the convolutional LSTM is established in Appendix B.1, with the end-to-end equivariance argument given in Section 4.3. Details on data, model architecture, and implementation are presented in Sections 4 and 5, and further elaborated in Appendices B.2 and C.1. Comprehensive descriptions of the training and evaluation procedures, including Rayleigh–Bénard simulation parameters, preprocessing steps, model and training hyperparameters, and evaluation metrics, are provided in Section 5 and Appendix C.1.

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

# A   ADDITIONAL DETAILS ON THE RAYLEIGH–BÉNARD SYSTEM AND ITS SYMMETRIES

## A.1   BOUSSINESQ APPROXIMATION AND NUMERICAL SOLUTION

For our simulation, we solve the so-called Boussinesq approximation of the compressible Navier-Stokes equations, where the fluid evolves according to the incompressible Navier-Stokes equations (continuity (3) and momentum (4)), even though the system is ultimately driven by an inhomogeneous density. Instead, the buoyancy force is modeled by a one-directional coupling with the energy conservation equation (5). In summary, we solve the following system of coupled PDEs in a rectangular domain $\Omega = (0, 2\pi) \times (0, 2\pi) \times (-1, 1)$ with periodic boundary conditions in the horizontal (i.e., first and second) directions and standard fixed walls boundary conditions with constant temperatures at the bottom and the top.

$$\nabla \cdot u = 0, \tag{3}$$

$$\frac{\partial u}{\partial t} + (u \cdot \nabla)u = \nabla p + \sqrt{\frac{Pr}{Ra}}\Delta u + Te_3, \tag{4}$$

$$\frac{\partial T}{\partial t} + u \cdot \nabla T = \frac{1}{\sqrt{RaPr}}\Delta T. \tag{5}$$

Here, $u(x, t) \in \mathbb{R}^3$ is the three-dimensional velocity (depending on space $x \in \Omega \subset \mathbb{R}^3$ and time $t \in \mathbb{R}^{\geq 0}$) and the scalar fields $p(x, t)$ and $T(x, t)$ denote the pressure and temperature, respectively. The canonical unit vector in the vertical direction is denoted as $e_3$. The dimensionless numbers $Ra$ and $Pr$ are the Rayleigh and Prandtl numbers, respectively, cf. Pandey et al. (2018) for a more detailed description. The Prandtl number is a constant depending on the fluid properties (kinematic viscosity divided by thermal diffusivity), and we use the common value $Pr = 0.7$. We will study different values of $Ra$, which is the main influence factor in terms of the system complexity. For the numerical simulations to generate our training data, we have used the Julia code `oceananigans.jl` (Wagner et al., 2025), which introduces a finite volume discretization in the form of a grid with $N = N_1 \times N_2 \times N_3 = 48 \times 48 \times 32$ elements in space. These are equidistantly placed in all three directions so that we have a grid size of $\delta_1 = \delta_2 = 2\pi/48$ and $\delta_3 = 2/32$.

## A.2   SYMMETRIES IN THE RAYLEIGH-BÉNARD SYSTEM

The PDE we consider is a modification of the Navier-Stokes equations (NSE) with an additional temperature-dependent buoyancy term in the vertical direction. It is well-known that the 3D NSE are equivariant under actions of the symmetry group E(3) (Olver, 1993; Wang et al., 2021). Consequently, our system inherits the translational equivariance and $O(2)$ equivariance in the two horizontal directions, as only the vertical component of the momentum equation is altered. In addition, the translational symmetry in the vertical direction is broken by the fixed-temperature boundary conditions at the bottom and top. Locally, however, and sufficiently far away from the walls, equivariance should approximately hold, which we will exploit in our architecture.

The symmetries can be formalized as equivariance with respect to three-dimensional rigid transformations $R(x) = Ax + b$, where $A$ leaves the vertical direction invariant, i.e., it has a $2 \times 2$ orthogonal block in the top-left corner and acts like the identity in the third (= the vertical) dimension. Together

with function composition as the operation, the set of these rigid transformations yields a group $(G, \circ)$.

An action of $G$ on scalar fields is simply defined via function composition, that is, by rotating or translating the underlying coordinate domain. This action is extended to vector- or tensor-fields, but here we have to transform their components too:

- For a scalar field $s : \mathbb{R}^3 \to \mathbb{R}$, a group action is defined by $R \cdot s = s \circ R$.
- For a vector field $v : \mathbb{R}^3 \to \mathbb{R}^3$, a group action is defined by $R \cdot v = A^\top v \circ R$.
- For a tensor field $M : \mathbb{R}^3 \to \mathbb{R}^{3 \times 3}$, a group action is defined by $R \cdot M = A^\top M A \circ R$.

From these definitions, equivariance of the Rayleigh-Bénard convection holds in the following sense: Writing equations (4) and (5) as $u_t = f(u, T, p)$ and $T_t = g(u, T, p)$, it holds that

$$u_t = R^{-1} \cdot f(R \cdot u, R \cdot T, R \cdot p) \quad \text{and} \quad T_t = R^{-1} \cdot g(R \cdot u, R \cdot T, R \cdot p). \tag{6}$$

*Proof.* We have

$$u_t = f(u, T, p) = -(\nabla u)^\top u + \nabla p + \sqrt{\frac{Pr}{Ra}} \Delta u + T e_3,$$

$$T_t = g(u, T, p) = -(\nabla T)^\top u + \frac{1}{\sqrt{RaPr}} \Delta T.$$

To treat vector fields consistently in column format, we have written $u \cdot \nabla u$ in (4) and $u \cdot \nabla T$ in (5) as $(\nabla u)^\top u$ and $(\nabla T)^\top u$, respectively.

We state the following vector calculus identities without proof, but note that they follow from straightforward calculations:

$$\nabla(R \cdot w) = R \cdot \nabla w, \qquad \text{(for } w \text{ a scalar or vector field)} \tag{7}$$
$$\nabla \cdot (R \cdot w) = R \cdot (\nabla \cdot w), \qquad \text{(for } w \text{ a vector or tensor field)} \tag{8}$$
$$(R \cdot w)^\top (R \cdot v) = R \cdot (w^\top v), \qquad \text{(for } w, v \text{ vector or tensor fields)} \tag{9}$$

Both $f$ and $g$ can be decomposed as linear combinations of

$$(\nabla v)^\top u, \quad \Delta v, \quad \nabla p, \quad \text{and} \quad T e_3, \tag{10}$$

where $v \in \{T, u\}$. Since $R$'s action is linear, it suffices to show equivariance for these terms individually, i.e.,

$$(\nabla(R \cdot v))^\top (R \cdot u) = R \cdot (\nabla v)^\top u, \tag{11}$$
$$\Delta(R \cdot v) = R \cdot \Delta v, \tag{12}$$
$$\nabla(R \cdot p) = R \cdot \nabla p, \quad \text{and} \tag{13}$$
$$(R \cdot T)e_3 = R \cdot (T e_3). \tag{14}$$

For (11), we apply first (7) and then (9). For (12), note that $\Delta v = \nabla \cdot \nabla v$, and one can thus use first (7) and then (8). Equation (13) follows from (7). Finally, we have for (9),

$$(R \cdot T)e_3 = (T \circ R)e_3 = T e_3 \circ R = A^\top T e_3 \circ R = R \cdot (T e_3), \tag{15}$$

since $A$ leaves $e_3$ invariant. Therefore, we have

$$f(R \cdot u, R \cdot T, R \cdot p) = R \cdot f(u, T, p) \quad \text{and} \quad g(R \cdot u, R \cdot T, R \cdot p) = R \cdot g(u, T, p).$$

Equation (6) then follows. $\qquad \square$

### A.3 ADDITIONAL SYSTEMS WITH BROKEN E(3) SYMMETRY

Even though the paper is concerned with Rayleigh Bénard convection only, there exists a multitude of systems where our architecture can be of use. Examples include, but are not limited to:

- systems with directed forces, such as

- buoyancy forces in the Rayleigh-Taylor instability (see Ohana et al. (2024) and the related website `https://polymathic-ai.org/the_well/datasets/rayleigh_taylor_instability` for an exemplary video).
    - magnetic fields in magneto-hydrodynamics.
- systems with symmetry-breaking flow structures such as
    - the flow through pipelines, where the symmetry is broken along the flow direction, leading to O(2) symmetry in the radial direction and a shift equivariance along the transport direction.
    - jet flows such as the exhaust gas coming out of a jet engine, where the symmetry is broken along the jet direction. As the jet increases in diameter in the downstream direction, the remaining symmetry would be O(2) around the jet center axis.

## B  ARCHITECTURAL AND IMPLEMENTATION DETAILS OF THE SURROGATE AND BASELINES

### B.1  LSTM EQUATIONS AND EQUIVARIANCE

Our convolutional LSTM architecture is defined as follows:

$$i_t = \sigma \left( z_t * \psi^{zi} + h_{t-1} * \psi^{hi} + c_{t-1} * \psi^{ci} + b^i \right)$$
$$f_t = \sigma \left( z_t * \psi^{zf} + h_{t-1} * \psi^{hf} + c_{t-1} * \psi^{cf} + b^f \right)$$
$$c_t = f_t \odot c_{t-1} + i_t \odot \tanh \left( z_t * \psi^{zc} + h_{t-1} * \psi^{hc} + b^c \right)$$
$$o_t = \sigma \left( z_t * \psi^{zo} + h_{t-1} * \psi^{ho} + c_t * \psi^{co} + b^o \right)$$
$$h_t = o_t \odot \tanh \left( c_t \right)$$

Here, we use peephole connections, where the cell state is included in the gates to control the information entering and leaving the cell state more accurately. Replacing the Hadamard products of traditional peephole connections with convolutions ensures that the LSTM remains equivariant, as shown below. Notably, this approach also resolves the issue of exploding gradients we initially observed in our experiments when using a standard 3D convolutional LSTM without convolutional peephole connections.

The new latent representation $z_{t+1}$ is predicted based on the current hidden state via

$$z_{t+1} = z_t + h_t * \psi + b.$$

The output $z_{t+1}$ of the current time step is then used as the input for the next time step, allowing the model to autoregressively forecast future states.

**Equivariance of the LSTM system dynamics.** The LSTM can be seen as modeling a dynamical system of the form

$$(z_t, c_{t-1}, h_{t-1}) \mapsto (z_{t+1}, c_t, h_t), \tag{16}$$

where the remaining variables $i_t, f_t$ and $o_t$ can be computed solely from $z_t, c_{t-1}, h_{t-1}$. Equivariance in this context means that a transformed input yields a transformed output, i.e.,

$$(\rho(g)z_t, \rho(g)c_{t-1}, \rho(g)h_{t-1}) \mapsto (\rho(g)z_{t+1}, \rho(g)c_t, \rho(g)h_t). \tag{17}$$

It follows from three facts, all due to Weiler & Cesa (2019): Firstly, convolutions are equivariant in the sense that $\rho(g)\ell * \psi = \rho(g)(\ell * \psi)$ for an arbitrary feature map $\ell$. Secondly, the action is linear, i.e., $\rho(g)\ell + \rho(g)\ell' = \rho(g)(\ell + \ell')$. And thirdly, it satisfies $\sigma(\rho(g)\ell) = \rho(g)\sigma(\ell)$ whenever $\sigma$ is a pointwise function. Since all feature maps are transformed jointly, the Hadamard product is equivariant too, i.e., $\rho(g)\ell \odot \rho(g)\ell' = \rho(g)(\ell \odot \ell')$.

### B.2  IMPLEMENTATION DETAILS

The implementation of 3D convolutions with height-dependent kernels and local vertical parameter sharing, as introduced in Section 4.1.2, can be efficiently realized by wrapping 2D convolution operations, avoiding the need for custom CUDA kernels and ensuring compatibility with existing deep learning frameworks like PyTorch or TensorFlow.

We consider feature maps $f : \mathbb{R}^3 \to \mathbb{R}^C$ represented as 5D tensors of shape $B \times C \times N_1 \times N_2 \times N_3$, where $B$ is the batch size and $N = N_1 \times N_2 \times N_3$ defines the spatial resolution.

**3D convolutions with height-dependent kernels.** To apply separate kernels for each of the $H_{\mathsf{out}}$ vertical positions in the output feature map, we extract vertical receptive fields of size $h$ from the input tensor. Each field has shape $B \times C_{\mathsf{in}} \times N_1 \times N_2 \times h$. These are sliced per target output height and rearranged by merging the input channels and vertical dimensions, resulting in a tensor of shape $B \times (h \cdot C_{\mathsf{in}}) \times N_1 \times N_2$. We then apply a 2D convolution with $h \cdot C_{\mathsf{in}}$ input channels and $C_{\mathsf{out}}$ output channels to compute a single output slice at one height.

To parallelize this across all $H_{\mathsf{out}}$ heights, we stack the receptive fields along the channel dimension, producing a tensor of shape $B \times (H_{\mathsf{out}} \cdot h \cdot C_{\mathsf{in}}) \times N_1 \times N_2$, and then apply a grouped 2D convolution, dividing the channels into $H_{\mathsf{out}}$ groups, each corresponding to a separate height (not to be confused with group-equivariant convolutions (Cohen & Welling, 2016)). After processing, the output tensor is reshaped back to $B \times C_{\mathsf{out}} \times N_1' \times N_2' \times H_{\mathsf{out}}$.

**Local vertical parameter sharing.** To share kernels across neighboring vertical positions, we define a vertical neighborhood of size $k$, such that each kernel is applied across heights $\{x_3 - k, \ldots, x_3 + k\}$, yielding a total of $n = 2k + 1$ receptive fields, cf. Fig. 3 for a sketch. For each output height, we collect all $n$ local receptive fields and apply the same kernel across them.

To handle boundary effects, we learn an additional set of $k$ kernels both for the top and bottom of the domain, ensuring that each vertical position—regardless of its location—receives the same number of kernel applications.

After convolution, the features from all kernels applied to a vertical position are concatenated along the channel dimension, resulting in $C_{\mathsf{out}} = n \cdot C$ output channels, where $C$ is the number of kernels learned per height. All of this can be parallelized by stacking all vertical neighborhoods along the batch dimension and applying a single grouped 2D convolution as described above.

### B.3 BASELINE ARCHITECTURES

In addition to our proposed equivariant surrogate, we implemented two widely used baseline architectures, 3D U-Nets and 3D Fourier Neural Operators (FNOs). Their configurations are detailed below.

**3D U-Net.** We implement a 3D U-Net with four downsampling and four upsampling stages and skip connections between the encoder and decoder. Each block uses two 3D convolutions with a kernel size of three and ReLU nonlinearities. Downsampling is performed via max pooling; upsampling uses trilinear interpolation. The final layer is a $1 \times 1 \times 1$ convolution to the target channels. We set the base width to $C_0 = 28$ hidden channels, with channels being doubled/halved after each downsampling/upsampling step, respectively, yielding 7.7M parameters.

**3D Fourier Neural Operator (FNO).** We use the `neuralop` (Kovachki et al., 2023; Kossaifi et al., 2024) 3D FNO implementation with 16 Fourier modes, 4 spectral layers, 20 hidden channels, GELU nonlinearities, and 7.4M parameters. Each FNO layer applies a spectral convolution in Fourier space (truncating to the specified modes) plus a pointwise linear transform in physical space. We do not use dropout and apply an $L_2$ weight decay of $1 \times 10^{-4}$ during training.

## C TRAINING AND METHODOLOGICAL CONSIDERATIONS

### C.1 TRAINING DETAILS AND COMPUTING RESOURCES

Both models were trained using the Adam optimizer (Kingma & Ba, 2017) with an MSE loss, a batch size of 64, a learning rate of $1 \times 10^{-3}$, and learning rate decay. Training was stopped after convergence of the validation loss. Dropout, $D_4$ data augmentation, and batch normalization (which was applied only to the autoencoder) were used. The LSTM was trained using backpropagation through time and scheduled sampling (Bengio et al., 2015) for gradual transition from teacher forcing[4] to autoregressive predictions.

---

[4]Teacher forcing uses the ground truth as the LSTM input during training, instead of its previous output.

Training used a total of 3400 GPU hours on NVIDIA A100 GPUs, with 1200 GPU hours dedicated to the models for final evaluation. Hyperparameters were manually tuned due to the long training times. Training the non-equivariant and equivariant autoencoders took approximately 8.2 and 13.7 hours, respectively, averaged over three runs. Notably, the equivariant autoencoder reached the non-equivariant model's final validation loss after only 1.3 hours. The corresponding LSTM models required 33.9 hours for the non-equivariant and 36.4 hours for the equivariant one. At inference time, the equivariant model has a 22% overhead compared to the non-equivariant one.

**Training times at different autoregressive training horizons.** To quantify the efficiency gap between our approach and the baselines, we also report average training times per epoch for the $D_4$-steerable LSTM, FNO, and U-Net. As shown in Table 1, the computational cost of FNOs and U-Nets grows dramatically with longer training horizons, whereas our model remains efficient because forecasting is performed entirely within the latent space. This motivates training our models with 50 autoregressive steps, while FNO and U-Net baselines were restricted to 5 steps (see Section 5).

Table 1: **Training times per epoch (in seconds) for different architectures.** Compared to FNOs and U-Nets, our latent-space $D_4$-steerable model trains an order of magnitude faster when using longer autoregressive horizons, since forecasting is performed directly in latent space. This highlights its scalability for efficient long-term forecasting.

| Training steps | $D_4$-steerable LSTM (ours) [s] | FNO [s] | U-Net [s] |
|---|---|---|---|
| 5 | 582 | 916 | 1091 |
| 50 | 1310 | 8876 | 12 939 |

## C.2 DISCUSSION ON TWO-STEP TRAINING APPROACH

In this work, we adopted a two-step training strategy in which the autoencoder (CAE) and the LSTM are trained separately. While end-to-end training may appear more natural, our experiments showed that this approach offers several important advantages. Below, we provide a detailed discussion of this design choice, including the challenges we observed with end-to-end training and possible future improvements.

**Challenges of end-to-end training**

- **Mixing of compression and forecasting.** When trained jointly, the LSTM can attempt to correct errors made by the CAE. This leads to unstable behavior in long-horizon autoregressive forecasts, as forecasting and reconstruction tasks interfere with each other.
- **High computational cost.** End-to-end training requires backpropagation through the encoder, LSTM, and decoder over time, making it both computationally expensive and memory-intensive.

One possible mitigation is to decode the LSTM's prediction, re-encode it, and feed it back into the LSTM for each autoregressive step. However, this introduces extreme inefficiency. On the other hand, separate training has multiple advantages.

**Advantages of separate training**

- **Clear separation of tasks.** In end-to-end training, the model tends to integrate forecasting logic into the autoencoder, which is especially harmful for autoregressive forecasting in latent space over long horizons. By contrast, training the CAE and LSTM separately ensures that the CAE focuses purely on compression and reconstruction, while the LSTM is specialized for temporal dynamics.
- **Efficiency.** Training the LSTM on pre-computed latent representations allows the forecasting loss to be computed directly in latent space, without backpropagating through the decoder and encoder. This reduced training time per epoch by approximately a factor of 20 (0.36 hours vs. 7.52 hours per epoch in our experiments), while also significantly lowering memory requirements.
- **Dynamical systems perspective.** From a modeling point of view, the two-step approach aligns with the idea of first learning a low-dimensional manifold that represents the system

dynamics (Connor & Rozell, 2020), and then training a forecasting model restricted to that manifold.

We believe that separate training could be enhanced by a careful end-to-end fine-tuning step. This would allow the latent representation to adapt to the forecasting task while retaining the benefits of efficient pre-training. Such an approach, however, requires a careful balancing of reconstruction and forecasting losses, as well as extensive hyperparameter tuning.

### C.3    ADVANTAGES OVER CLASSICAL PDE SOLVERS

A key motivation for surrogate modeling is to overcome the large computational cost of classical PDE solvers, particularly for high-dimensional turbulent systems such as Rayleigh–Bénard convection. In this appendix, we provide additional measurements and discuss the advantages of our approach compared to numerical solvers.

**Multi-query capability.** Classical solvers must integrate each trajectory individually, which becomes prohibitively expensive in scenarios such as uncertainty quantification, optimization, or control. By contrast, the surrogate model can efficiently predict large batches in parallel, enabling applications that are infeasible with traditional PDE solvers.

**Inference speed.** We benchmarked our surrogate model against the highly optimized solver `oceananigans.jl`. For the Rayleigh number considered in this work, the surrogate already shows a modest speedup while maintaining high accuracy. More importantly, the computational cost of classical solvers grows rapidly with increasing complexity, such that scaling towards more turbulent regimes will strongly favor machine learning approaches.

Table 2: **Average inference times** (500-step forecast) of our surrogate model ($D_4$-steerable CAE+LSTM) compared to the PDE solver `oceananigans.jl`. Reported times include only the forecasting computation (no memory or SSD I/O for saving the output). While single-trajectory runtimes are comparable, the surrogate provides dramatic speedups in multi-query settings, where large batches can be simulated in parallel.

| Scenario | **Ours** [s] | **Solver** [s] | **Speedup** |
|---|---|---|---|
| Single trajectory, full sequence output | 18.38 | 28.34 | 1.5× |
| Single trajectory, only final state output | 12.06 | 25.31 | 2.1× |
| 256 trajectories, full sequence output | 257.12 ($\approx$1.00 / sim.) | — | 28.3× |
| 256 trajectories, only final state output | 47.05 ($\approx$0.18 / sim.) | — | 140.6× |

Table 2 summarizes the average speedup factors for computation (CAE+LSTM vs. `oceananigans.jl`). The results highlight the clear advantage of surrogate models in multi-query settings, where large ensembles of trajectories can be simulated efficiently in parallel. In addition, our approach provides a substantial computational benefit when only the final state is required, since forecasting can be performed entirely in latent space and only the last state needs to be decoded.

**Differentiability.** Another strength of neural surrogates is their differentiability. Unlike most classical numerical solvers, which are either not differentiable (in particular high-performance solvers for very large simulations and commercial codes) or require the tedious implementation of adjoint equations,[5] our model can be directly differentiated end-to-end. This enables gradient-based optimization, data assimilation, control, and parameter identification. We believe that the combination of scalability, efficiency, and differentiability makes surrogate models particularly attractive for downstream tasks in scientific machine learning.

---

[5]It should be noted that there have been various attempts in the recent past to design fully differentiable solvers, in particular using the backpropagation functionalities included in modern ML packages such as PyTorch; cf., e.g., Holl & Thuerey (2024); Franz & Thuerey (2024); Winchenbach & Thuerey (2025).

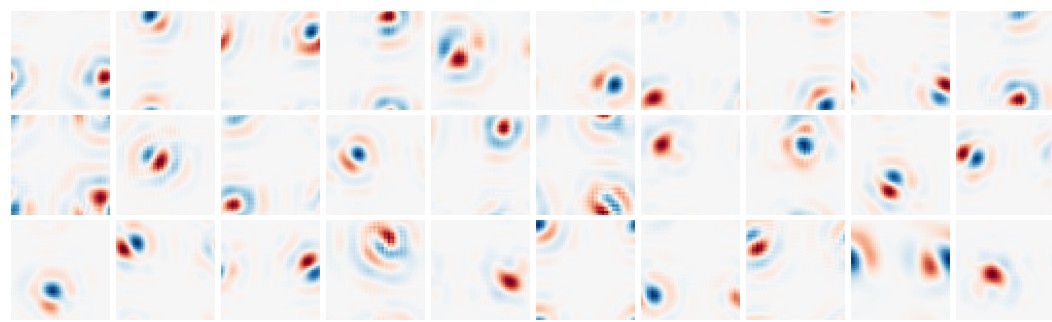

Figure 7: **Representative sensitivity patterns** of the temperature field (horizontal slices) associated with individual latent entries of the $D_4$-steerable autoencoder.

## D  ADDITIONAL EXPERIMENTS

### D.1  RAYLEIGH-TAYLOR INSTABILITY

Additional experiments are currently in progress, where we study the Rayleigh-Taylor instability. This is a system with two fluid layers of different densities stacked on top of each other, with the lower-density fluid at the bottom. Due to buoyancy, the layers start mixing, giving rise to highly complex patterns. The dataset consists of multiple trajectories with $128 \times 128 \times 128$ snapshots of the density field and three-dimensional velocity vector field. It is provided by the popular benchmark library "The Well" Ohana et al. (2024).

Due to the larger dataset size, the experiments are time consuming and thus, currently still ongoing with promising intermediate results. We will provide a detailed documentation and discussion in the final document, once the experiments are finished.

### D.2  LATENT SPACE STRUCTURE & VISUALIZATION

To further investigate the structure of the learned latent space, we analyzed the sensitivity of individual latent entries with respect to the input fields. For a given latent variable $i$, we computed

$$\frac{\partial z_i}{\partial s}$$

via backpropagation, and then averaged this quantity over a large set of snapshots. This yields a representative spatial pattern indicating which input-space structures most strongly activate each latent variable.

Figure 7 displays these patterns for the equivariant $D_4$-steerable autoencoder. Each panel shows the horizontal slice of maximal norm within the 3D pattern of the temperature field represented by one latent entry. The patterns exhibit localized oscillatory structures with distinct orientation and spatial frequency, closely resembling classical Gabor filters.

Furthermore, Figure 8 highlights a striking property of the equivariant model: for a single latent feature, all eight elements of the $D_4$ group—$\{e, r, r^2, r^3, \kappa, r\kappa, r^2\kappa, r^3\kappa\}$—appear explicitly in the learned representation. These rotated and reflected variants form an orbit under the $D_4$ action, confirming that the latent variables correctly respect and encode the underlying symmetry structure of the Rayleigh–Bénard system.

For completeness, we also performed the same analysis for the non-equivariant 3D autoencoder. We did not observe qualitative differences in the types of primitive spatial features learned, apart from the absence of the explicit $D_4$-structured organization. Thus, the primary distinction between the models is not the nature of the local filters themselves, but the equivariant model's structured arrangement of these filters into symmetry-consistent orbits.

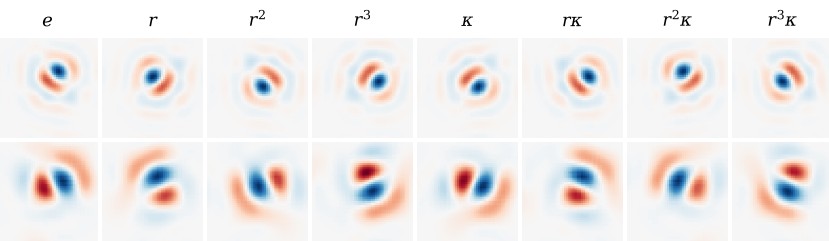

| $e$ | $r$ | $r^2$ | $r^3$ | $\kappa$ | $r\kappa$ | $r^2\kappa$ | $r^3\kappa$ |

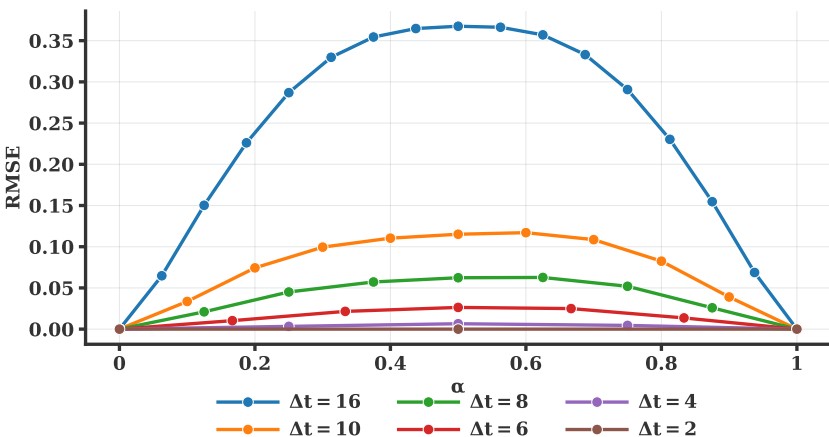

Figure 8: **Orbit of latent sensitivity patterns** under the $D_4$ group action. Each panel corresponds to a transformation in $\{e, r, r^2, r^3, \kappa, r\kappa, r^2\kappa, r^3\kappa\}$, composed out of rotations $r$ and reflections $\kappa$.

Figure 9: **Interpolation error (RMSE)** as a function of the interpolation coefficient $\alpha \in [0, 1]$ for several temporal gaps $\Delta t$. For small gaps ($\Delta t = 2, 4$), interpolation quality remains close to the underlying autoencoder reconstruction error, while larger gaps ($\Delta t = 6, 8, 10, 16$) lead to gradually increasing—but still smooth and non-catastrophic—errors. The symmetric error curves reflect the linear interpolation path between the endpoints.

### D.3 INTERPOLATION IN LATENT SPACE

To assess whether the latent space learned by our equivariant autoencoder admits meaningful interpolations between states, we conducted a series of experiments in which two snapshots $s_{t_0}$ and $s_{t_1}$ were encoded into latent representations $z_{t_0}$ and $z_{t_1}$, followed by linear interpolation in latent space,

$$z(\alpha) = (1 - \alpha)\, z_{t_0} + \alpha\, z_{t_1}, \qquad \alpha \in [0, 1],$$

and subsequent decoding into physical space. For a given interpolation range $\Delta t = t_1 - t_0$, we compared the decoded interpolants at intermediate $\alpha$ against the corresponding ground-truth snapshots.

Figure 9 summarizes the reconstruction errors obtained for several choices of $\Delta t$. Importantly, these errors *exclude* the intrinsic autoencoder reconstruction error of approximately $0.039$, which must be added to obtain the full error. As can be seen from the figure, interpolations over small temporal gaps exhibit errors essentially indistinguishable from the base reconstruction error, while larger gaps degrade gracefully. For example, for $\Delta t = 6$, the additional RMSE takes the values $0.01$, $0.022$, $0.026$, $0.025$, and $0.014$ for the five interpolated states. For a temporal gap of $\Delta t = 4$, the RMSE reduces to $0.003$, $0.007$, and $0.005$. When interpolating only a single intermediate step, the interpolation error becomes negligible.

Representative qualitative results for an interpolation range of $\Delta t = 10$ are shown in Figure 10. Across all cases, the interpolated snapshots remain physically plausible and match the overall plume and roll structures of the ground truth. Deviations are primarily small spatial displacements or mild damping of fine-scale features, consistent with the measured RMSE values.

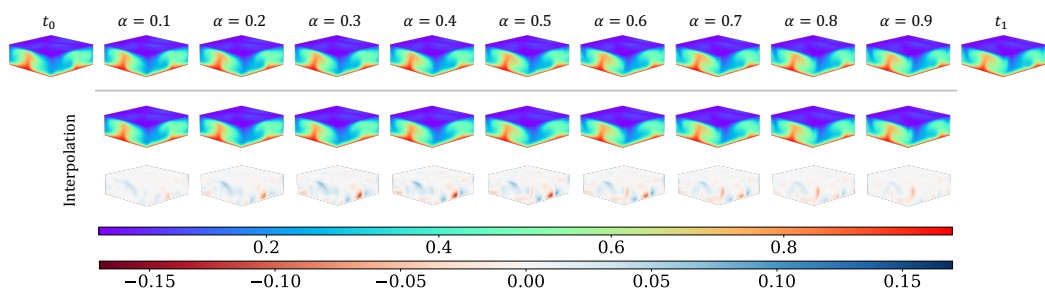

Figure 10: **Qualitative interpolation results** for a temporal gap of $\Delta t = 10$. Top row: ground-truth snapshots at times $t = 0$ through $t = 10$. Middle row: decoded latent interpolants at intermediate positions $\alpha = \frac{1}{10}, \ldots, \frac{9}{10}$. Bottom row: corresponding reconstruction differences. The interpolated states remain physically plausible and reproduce the large-scale flow structures of the reference snapshots, with discrepancies dominated by mild spatial misalignment and attenuation of fine-scale features, consistent with the measured interpolation RMSE.

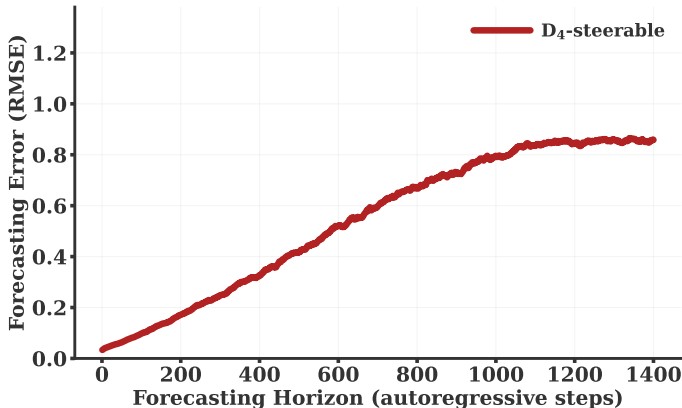

Figure 11: **Median RMSE over very long autoregressive rollouts** (0–1400 steps) for the $D_4$-steerable CAE+LSTM. The error grows approximately linearly up to ~1200 steps and then saturates, forming a plateau around RMSE $\approx 0.85$.

These results demonstrate that (i) the latent space learned by the equivariant $D_4$-steerable autoencoder retains an approximately linear structure over short temporal horizons, and (ii) the model allows interpolation of snapshots in a physically meaningful manner. In particular, measurements of the latent-space distance $\|z_t - z_{t+k}\|$ reveal an approximately linear increase of about 3.2 per time step for small $k$, further supporting this observation. Consequently, the model can be used to increase temporal resolution in sparsely sampled data by decoding latent interpolants.

### D.4 STABILITY UNDER VERY LONG AUTOREGRESSIVE ROLLOUTS

To evaluate the stability of our surrogate model over extended horizons, we perform autoregressive rollouts of up to 1400 steps using the $D_4$-steerable CAE+LSTM. The median RMSE increases approximately linearly during the first ~1200 steps and subsequently saturates, forming a plateau at RMSE $\approx 0.85$ (Fig. 11). This behavior reflects the gradual accumulation of prediction inaccuracies over time, while the model continues to generate coherent and physically plausible temperature fields. Even at $t = 1400$, the large-scale convective patterns remain well preserved; the main differences compared to the reference solution arise from mild spatial misalignment between the predicted and true plume locations (Fig. 12).

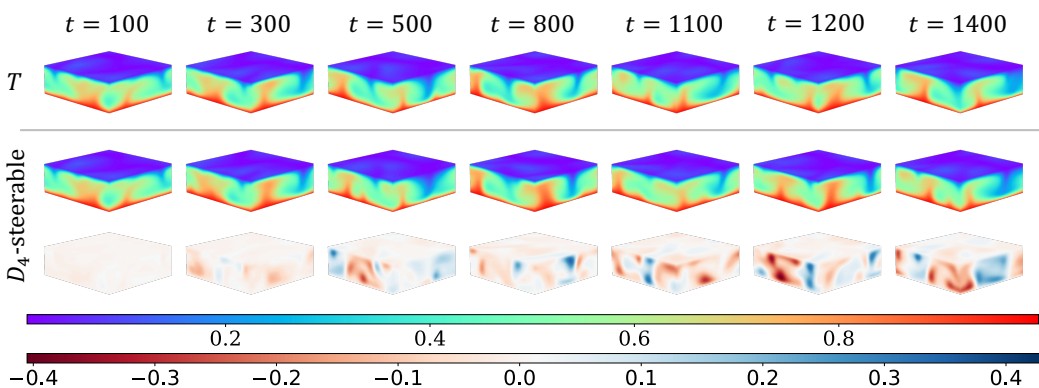

Figure 12: **Representative qualitative forecast up to t = 1400.** Top row: ground-truth temperature field $T$. Middle row: predictions of the $D_4$-steerable surrogate model. Bottom row: difference fields. Across all horizons, the model preserves the overall convective structure. At longer horizons, discrepancies consist primarily of slight spatial misalignment between predicted and true plume locations, rather than a breakdown of the underlying flow patterns, consistent with the saturated median RMSE of approximately 0.85.

### D.5 NOISE ROBUSTNESS OF THE $D_4$-EQUIVARIANT AUTOENCODER

In addition to the high-fidelity evaluation presented in the main paper, we investigate the robustness of our $D_4$-equivariant autoencoder when subjected to noisy input fields. Since practical data acquisition processes may introduce measurement noise, it is important to understand how reconstruction quality degrades under controlled perturbations and how this robustness can be improved within our framework.

**Setup.** We evaluate the trained $D_4$-equivariant autoencoder on temperature-velocity snapshots corrupted by additive Gaussian noise, $s_{\mathrm{noisy}} = s + \varepsilon$, with $\varepsilon \sim \mathcal{N}(0, \sigma^2)$. Noise levels range from $\sigma = 0.01$ to $\sigma = 0.2$ (in standardized units). We report the reconstruction RMSE between the decoder output on the noisy input and the clean reference snapshot.

To improve robustness, we additionally train a variant of the autoencoder using a *noise-consistency* regularizer, which encourages the encoder to map clean and noisy versions of the same snapshot to similar latent representations:

$$\mathcal{L}_{\mathrm{cons}} = \|\mathrm{Enc}(s) - \mathrm{Enc}(s + \varepsilon)\|_F^2, \quad \epsilon \sim \mathcal{N}(0, \sigma^2), \quad \sigma \sim U(0, 0.2).$$

This term is added to the reconstruction loss during training, using the same architecture and hyperparameters as the baseline $D_4$ model.

**Results.** Figure 13 shows the reconstruction RMSE as a function of noise level. The baseline model exhibits smooth, non-catastrophic degradation as noise increases, preserving reasonable reconstruction quality even at higher noise levels. Incorporating the noise-consistency regularizer substantially improves robustness: for $\sigma \leq 0.05$, reconstruction errors remain close to the clean-input baseline, and even for $\sigma = 0.1$–$0.2$, the consistency-regularized model achieves lower errors than the non-regularized variant. *These results are preliminary, and we expect that additional tuning of the regularization weight and training procedure will further improve the reconstruction RMSE of the consistency-regularized model.*

### D.6 SENSITIVITY OF HEIGHT-DEPENDENT KERNELS

To assess the robustness of the height-dependent convolutional design introduced in Section 4.1.2, we evaluate the $D_4$-steerable autoencoder under several architectural and spatial configurations. Specifically, we vary (i) the vertical kernel size and (ii) the vertical spatial resolution $N_3$. All experiments use the same total number of trainable parameters to ensure a fair comparison. Reconstruction quality is measured in terms of RMSE on the clean test set.

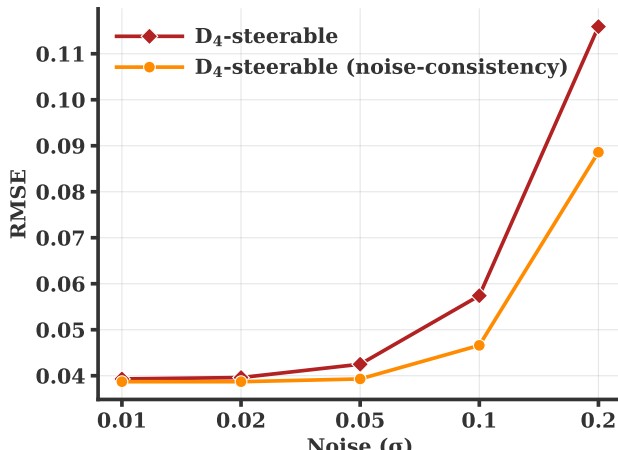

Figure 13: **Reconstruction RMSE under additive Gaussian noise** for the $D_4$-equivariant autoencoder and its noise-consistency variant. The consistency regularizer improves robustness across all tested noise levels, with particularly strong gains for $\sigma \leq 0.05$. The results are preliminary, and we expect further improvements with additional tuning.

**Varying vertical kernel size.** Table 3 summarizes the effect of changing the vertical kernel size while keeping the horizontal kernel size fixed at $(5, 5)$ with $N_3 = 32$. Performance remains stable across all tested configurations, with only marginal variation in RMSE.

Table 3: **Sensitivity to vertical kernel size** for the $D_4$-steerable autoencoder ($N_3 = 32$). The reconstruction error varies only slightly across the tested kernel depths.

| Kernel size | RMSE |
|---|---|
| $(5, 5, 3)$ | 0.0395 |
| $(5, 5, 5)$ | 0.0391 |
| $(5, 5, 7)$ | 0.0416 |

**Varying vertical spatial resolution.** Finally, we assess sensitivity to the vertical grid resolution. Increasing the resolution from $N_3 = 32$ to $N_3 = 48$ while keeping the number of parameters fixed leads to a small increase in RMSE (Table 4), which is expected given the increased input dimensionality with the same number of parameters.

Table 4: **Sensitivity to vertical spatial resolution** for the $D_4$-steerable autoencoder with fixed parameter count. Higher vertical resolution slightly increases reconstruction error due to the larger input size.

| Vertical resolution $N_3$ | RMSE |
|---|---|
| 32 | 0.0391 |
| 48 | 0.0430 |

**Summary.** Across all experiments, reconstruction errors remain within a narrow range, demonstrating that the height-dependent kernel design is robust to the choice of vertical kernel size and the vertical spatial resolution.

### D.7 EFFECTS OF LOCAL VERTICAL PARAMETER SHARING

To complement the experiments, we evaluate how different degrees of vertical parameter sharing affect reconstruction accuracy in the $D_4$-steerable autoencoder. Vertical sharing restricts subsets of kernels to be reused across neighboring heights, reducing the number of independent vertical

parameters while preserving the height-dependent structure of the convolutions. All configurations below use the same total number of trainable parameters; only the vertical sharing pattern is varied.

**Experimental setup.** We consider four variants: no sharing, and three increasingly shared configurations (small, medium, large). The encoder specifications below denote the size of the vertical neighborhood over which kernels are shared at each layer; the decoder mirrors this pattern. All results are reported at vertical resolution $N_3 = 32$.

Table 5: **Reconstruction accuracy under different levels of vertical parameter sharing** at $N_3 = 32$. All models use the same parameter budget. Performance varies only slightly across configurations, with medium and large sharing achieving the lowest errors.

| Vertical parameter sharing (encoder) | RMSE |
|---|---|
| None $(1, 1, 1, 1, 1)$ | 0.0391 |
| Small sharing $(5, 3, 3, 1, 1)$ | 0.0408 |
| Medium sharing $(7, 5, 5, 3, 3)$ | 0.0377 |
| Large sharing $(9, 7, 7, 5, 5)$ | 0.0373 |

**Effect of vertical sharing.** Across all configurations, reconstruction accuracy remains within a narrow range (cf. Table 5), indicating that the model is robust to the specific choice of sharing pattern. Notably, medium and large sharing yield slightly lower errors than both the no-sharing and small-sharing variants. These improvements, although modest, suggest that moderate vertical kernel reuse can be beneficial without degrading the model's ability to represent height-dependent structure.

**Higher vertical resolution.** We additionally investigate the effect of vertical parameter sharing at increased resolution ($N_3 = 48$). These experiments are ongoing, but preliminary results indicate that vertical sharing becomes more effective as resolution increases; early runs show substantially improved accuracy relative to the non-shared baseline at the same parameter count. This trend aligns with the intuition that higher-resolution height dimensions provide more opportunities for efficient reuse of vertically local kernels. More complete results will be provided as experiments are finished.

