# OpenReview forum: "Surrogate Modeling of 3D Rayleigh-Bénard Convection with Equivariant Autoencoders"
_ICLR.cc/2026/Conference — Submitted to ICLR 2026_

### Official Review · Reviewer_gjPD · 2025-10-23

**Soundness:** 3
**Presentation:** 3
**Contribution:** 2
**Rating:** 4
**Confidence:** 4

**Summary:**

The authors present a method to obtain a surrogate model of the 3D Rayleigh-Bénard convection equation which is equivariant over rotations and translations in the horizontal 2D direction. They use a encoder-process-decoder architecture: (1) encoder based on steerable kernels, (2) LSTM to predict rollout predictions whose MLPs are substituted with equivariant kernels, and (3) decoder with steerable kernels and trilinear interpolation upsampling. The composition of equivariant components make the whole model to be E(2) equivariant in the horizontal dimension and height-dependant learnable kernels in the vertical direction, where the symmetry is broken due to the buoyancy dynamics. The model is tested over the mentioned equation resulting in lower errors than the non-equivariant counterpart with significantly less parameters. A long-term forecasting test also shows that the method outperforms FNO and U-Net baselines.

**Strengths:**

* The compression ratio of the autoencoder is very high and the results outperform the baselines by a decent margin.
* The local vertical parameter sharing is a smart solution to the vertical loss of symmetriy.
* The paper treats equivariance with mathematical rigour and proofs.

**Weaknesses:**

* The paper focuses primarily on a single specific equation and lacks validation on more complex phenomena or dynamical scenarios. One would expect a method designed for equivariant modeling to demonstrate broader applicability to a variety of PDEs exhibiting such symmetries and symmetry-breaking, but this is not explored in the paper.
* The contribution of the paper is quite limited. Equivariant autoencoders have been extensively explored in the literature, and the replacement of LSTM forward networks with an equivariant version represents only a modest contribution. The use of height-dependent kernels, however, is an interesting aspect.
* The code is not ready for review.

**Questions:**

* Lines 324-325: Are all 900 timesteps within the interval [100, 1000] included in the training data? Does the model perform any extrapolation to snapshots outside this range?
* Section 3.2: The autoencoder is evaluated across multiple Rayleigh numbers, which is appropriate given their strong influence on the system’s dynamics. However, the long-term forecasting experiments appear to be conducted for a single Rayleigh number, though this is not explicitly stated. The results would be more convincing if a similar multi-Ra number analysis were included for the forecasting stage as well.
* Section 4.1.2: One of the main contributions of the paper is the introduction of height-dependent kernels, yet this aspect remains relatively underexplored. How sensitive are the results to the choice of kernel size or vertical resolution?
* Line 965: The text mentions that the non-equivariant model was trained with data augmentation. Can the authors clarify whether similar augmentations (rotations, translations) were also applied to the long-term forecasting baselines (FNO, U-Net)? This would help ensure a fair comparison and clarify whether the performance gap stems primarily from architectural equivariance or from differences in training data diversity.

**Details Of Ethics Concerns:**

I have no ethics concerns.

---

> ### Author Response · Authors · 2025-11-24
>
> We thank the reviewer for the thoughtful and helpful comments. We conducted multiple new experiments in response, and the global comment summarizes the changes reflected in the revised PDF.
>
> Please find a list of detailed point-by-point replies to the identified weaknesses and open questions.
>
> ### Weaknesses
> 1. We appreciate this concern. Although the paper focuses on Rayleigh–Bénard convection, this system is already a challenging 3D turbulent PDE with strong nonlinearity and multi-scale dynamics; our experiments across different Rayleigh numbers (Fig. 4) also show that the benefits of equivariance increase with system complexity. Importantly, the proposed architecture is not specific to RBC: it applies directly—**without any architectural changes**—to any PDE with the same horizontal symmetry structure, as detailed in Appendix A.3. In line with the reviewer’s suggestion, we have begun evaluating the method on a second system, the 128×128×128 Rayleigh–Taylor instability, which is significantly more complex; these large-scale experiments are ongoing with promising intermediate results and will be documented in the final version.
> 2. While individual components (equivariant AEs, convLSTMs) exist, their combination in a fully 3D, symmetry-matched surrogate is, to our knowledge, new and substantially more challenging than 1D/2D settings. Our pipeline provides (i) a horizontally $E(2)$-equivariant 3D AE with a 3D latent space, (ii) height-dependent steerable kernels for broken vertical symmetry, and (iii) local vertical sharing enabling tractable 3D equivariant models. This design supports long-horizon latent forecasting (50 steps vs. 5 for full-resolution baselines) and yields strong empirical gains (e.g., $>98\%$ compression with lower RMSE).
> 3. An anonymized code repository is now available, including the equivariant CAE, convLSTM, height-dependent kernels, training/evaluation scripts, preprocessing code, and reproduction instructions.
>
> ### Questions:
> 1. Thank you for the opportunity to clarify this point. We use two datasets: training on $t\in[100,300]$ ($\Delta t=0.5$) and long-term forecasting evaluation on $t\in[100,1000]$. Long-horizon scores are computed by sliding 500-step autoregressive rollouts across all subsequences of the evaluation set. Thus, evaluation spans the full $[100,1000]$ range and requires genuine long-horizon extrapolation far beyond the training window.
> 2. Forecasting experiments are conducted at $\mathrm{Ra}=2500$, which we will make explicit. Our multi-$Ra$ autoencoder study (Fig. 4) shows that the advantage of the equivariant representation actually increases with higher $Ra$. Since the long-term forecasting performance in Sec. 5.1 is driven mainly by the quality and structure of this latent space, we expect the same trend to hold when training forecasters at other $Ra$ values. The forecasting module itself requires no architectural changes and can be trained independently for any $Ra$ in exactly the same way.
> 3. We have added a detailed analysis of this question in Appendix D.6 (kernel size and vertical resolution) and Appendix D.7 (vertical parameter sharing). Varying the vertical kernel depth and the vertical grid resolution shows that reconstruction errors remain within a narrow range, indicating that the height-dependent kernels are robust to these choices. We also evaluate several degrees of local vertical parameter sharing, which preserves height-dependent structure while reducing parameters. At the standard resolution ($N_3=32$), all variants achieve similar accuracy. Preliminary experiments at higher vertical resolution ($N_3=48$) show that sharing becomes significantly more beneficial in regimes of higher vertical resolution; these runs are ongoing, and full results will be included in a revised appendix. Overall, the height-dependent design and sharing scheme behave stably and are not sensitive to hyperparameters.
> 4. Thank you for raising this clarification. Yes, all long-term forecasting baselines (FNO, U-Net) were trained with the same $D_4$-consistent data augmentations (rotations and reflections) as our non-equivariant CAE. This ensures that every model receives identical symmetry information through the data, so the observed performance gap reflects the benefits of architectural equivariance and latent-space forecasting rather than differences in data diversity.

---

### Official Review · Reviewer_Uzfy · 2025-10-29

**Soundness:** 3
**Presentation:** 3
**Contribution:** 2
**Rating:** 6
**Confidence:** 3

**Summary:**

The paper designs an equivariant neural architecture for learning the phenomenon of buoyancy-driven fluid flow between a heated bottom and a cooled top plate involved in climate simulation, incorporating equivariant properties of the corresponding PDE solution directly inside the neural architecture. The authors validate this approach by comparing it to non equivariant surrogate models, FNO and U-Net, on the learning problem at hand.

**Strengths:**

- The paper introduce a novel architecture component by tailoring equivariant layers to their test case, with several innovating technical contributions.
- The presented architecture, $D_4$- steerable, outperforms other baselines with far fewer model paramaters for both short and long time horizons.

**Weaknesses:**

**W1** The obtained architecture seems to be applicable to more diverse test cases that the one presented: the paper would benefit from experiments on other datasets, all the more since the presented dataset is not standard and of moderate scale.

**Questions:**

**Q1** In what other types of simulation could the equivariant CNN be useful?

**Q2** l.321: "We generated a dataset of 100 randomly initialized 3D Rayleigh-Bénard convection simulations with Ra = 2500 and P r = 0.7," Are you sampling only initial conditions ? If Ra is set to 2500, how can different value of Ra be tested in Figure 4 (rightmost plot) ?

---

> ### Author Response · Authors · 2025-11-24
>
> We thank the reviewer for the thoughtful and helpful comments. We conducted multiple new experiments in response, and the global comment summarizes the changes reflected in the revised PDF.
>
> Please find a list of detailed point-by-point replies to the identified weaknesses and open questions.
>
> ### Weaknesses
> 1. We appreciate the suggestion to include additional test cases. Although our dataset is not a standard ML benchmark, 3D Rayleigh–Bénard convection at high Rayleigh numbers is already a challenging, fully turbulent system with a large state dimension and long temporal sequences; our setup follows standard configurations used in fluid dynamics and climate modeling. Importantly, the proposed architecture is directly applicable to other systems with the same symmetry structure—such as Rayleigh–Taylor instability—**without any modifications** to the architecture (see Appendix A.3). In line with the reviewer’s comment, we have begun evaluating our model on the $128\times128\times128$ Rayleigh–Taylor data from "The Well" [1]; due to their computational requirements these large-scale experiments are still running but show promising intermediate results, and we will provide full documentation in the final version.
>
> ### Questions
> 1. The proposed architecture is not specific to Rayleigh–Bénard convection; it applies directly—without any architectural changes—to any 3D system that shares the same symmetry structure (horizontal rotations/reflections, broken vertical symmetry). As outlined in Appendix A.3, this includes, for example, Rayleigh–Taylor instability, magneto-hydrodynamic flows, and driven pipe or channel flows. In this sense, the model provides a general template for building symmetry-matched 3D surrogates across a range of PDE systems.
> 2. Thank you for pointing out this ambiguity. For Fig. 4 (right), we retrained the autoencoder separately for each Rayleigh number, using datasets generated at the corresponding Ra value. The sentence cited in line 321 refers only to the Ra = 2500 case. We will clarify this in the revised manuscript. Notably, the trend in Fig. 4, where the equivariant model’s advantage increases at higher $Ra$, also indicates that the benefits extend to more turbulent regimes.
>
> [1] R. Ohana et al. The Well: a Large-Scale Collection of Diverse Physics Simulations for Machine Learning. NeurIPS Datasets and Benchmarks Track, 2024.

---

### Official Review · Reviewer_V5wy · 2025-10-31

**Soundness:** 3
**Presentation:** 3
**Contribution:** 3
**Rating:** 6
**Confidence:** 3

**Summary:**

This paper presents a surrogate modeling approach for three-dimensional Rayleigh-Bénard convection (RBC) using deep learning methods that respect spatial symmetries. The authors propose a two-stage architecture consisting of an equivariant convolutional autoencoder (CAE) and an equivariant convolutional LSTM. The CAE compresses high-dimensional flow fields into a structured latent space, while preserving horizontal rotational and reflectional symmetries through D4-steerable convolutions. The latent dynamics are then modeled with a convolutional LSTM that predicts temporal evolution in this compressed space. The model aims to approximate the underlying partial differential equations governing the RBC system, reducing computational cost relative to full numerical simulation.

The model operates on three-dimensional fields that contain temperature and velocity information across space. These fields are first passed through an encoder, which compresses them into a lower-dimensional latent representation while preserving spatial structure. A decoder then reconstructs the original field from this compressed form. For modeling time evolution, a recurrent neural network, specifically a convolutional LSTM, is used to predict future latent states based on previous ones. The system is trained in two separate phases: first, the autoencoder is optimized to minimize reconstruction error between the input and the decoded output; second, the LSTM is trained to forecast latent states over time. The full pipeline is evaluated using data from high-fidelity simulations of Rayleigh-Bénard convection at different Rayleigh numbers. According to the results, the proposed method achieves better data efficiency, lower reconstruction and forecasting errors, and requires fewer parameters than both standard convolutional neural networks and Fourier Neural Operator baselines.

**Strengths:**

The methodology is technically consistent and leverages symmetry-aware design principles grounded in group representation theory. The explicit use of D4-steerable convolutions ensures equivariance to horizontal rotations and reflections, which is physically appropriate for Rayleigh-Bénard convection where boundary conditions break vertical but not horizontal symmetry. The decision to apply height-dependent kernels is a reasonable approximation to accommodate vertical heterogeneity in the flow. The decomposition of spatial compression and temporal evolution into separate modules (CAE and LSTM) leads to a well-structured and interpretable model pipeline. This separation also reduces computational complexity, as the temporal predictor operates on latent tensors rather than full-resolution fields.

Empirical evaluation includes relevant comparisons with non-equivariant models and established operator learning baselines. The reported improvement in reconstruction error (approximately 40\% reduction in RMSE) and parameter efficiency (roughly an order of magnitude fewer parameters) is quantitatively clear. The experiments also cover different Rayleigh numbers, providing some evidence that the model generalizes to flows of varying complexity. The framework demonstrates that enforcing group-theoretic structure can lead to better inductive bias for spatiotemporal PDE systems.

**Weaknesses:**

The experimental validation remains limited in scope and does not fully establish the method’s robustness. All results are obtained on a single physical system (Rayleigh-Bénard convection) with idealized, noise-free data, which restricts the conclusions about general applicability. A joint end-to-end optimization might yield a more physically coherent latent space, though at higher cost. The study does not include ablations quantifying how much each design element (e.g., D4-steerable filters, local vertical parameter sharing) contributes to performance gains. Additionally, while the model is termed “equivariant,” it is only partially so: symmetry constraints are applied in horizontal planes but not in the vertical dimension, meaning the full $E(3)$ group is not represented. The reported efficiency gains are primarily relative to baseline models that do not exploit symmetries or that are trained under different regimes, which makes direct fairness of comparison uncertain. Finally, the paper does not explore stability or error accumulation over very long autoregressive rollouts, which is a key concern for temporal surrogate models.

**Questions:**

1. The model is trained in two separate phases: first the autoencoder, then the temporal predictor. While this improves training efficiency, it may introduce a mismatch between the latent representations learned by the encoder and those needed for accurate forecasting. Did the authors attempt any form of joint fine-tuning or end-to-end training, even partially? Can the authors comment on whether this decomposition leads to artifacts or long-horizon degradation in practice?

2. The proposed model architecture includes several components motivated by physical and architectural reasoning (e.g., D4-steerable convolutions, height-dependent filters, vertical parameter sharing). However, no ablation studies are provided to assess their relative contribution. Can the authors provide experiments or discussion that isolate the impact of these design choices?

3. Several baselines, such as U-Net and FNO, are included in the comparison. Were these models adapted for 3D input and trained with similar levels of supervision and data? FNO in particular is known to have strong performance in 2D PDE forecasting; was it adapted in a memory-efficient manner for 3D, or was the comparison constrained by hardware?

4. Does the learned latent space exhibit any physical interpretability? For instance, do certain channels correspond to flow structures such as rolls or plumes? Could latent vectors be interpolated or manipulated to generate physically meaningful transitions in the decoded space?

5. Although different Rayleigh numbers are used in training and evaluation, the setup assumes a fixed domain and set of boundary conditions. Would the model generalize across different physical configurations, such as changes in domain size, aspect ratio, or boundary heating profiles? If not, what would be required to make the surrogate model adaptive to such changes?

6. All training data appears to be generated from numerical simulations. How robust is the model to noise or distributional shifts, as would be expected in real-world experimental measurements or lower-fidelity simulations?

---

> ### Author Response · Authors · 2025-11-24
>
> We thank the reviewer for the thoughtful comments. We conducted several new experiments in response; the global comment summarizes the changes reflected in the revised PDF.
>
> Below, we provide point-by-point replies to the weaknesses and questions.
>
> ### Weaknesses
>
> 1. **Limited experimental scope:** Broader robustness is an important point. Although focused on 3D Rayleigh–Bénard convection, this is already a fully turbulent, high-dimensional benchmark. The architecture applies **without modification** to systems with the same symmetry. We have begun evaluating it on the $128\times128\times128$ Rayleigh–Taylor instability, and the ongoing runs show promising intermediate results; a detailed discussion will be included in the final version (App. D.1). Robustness was extended in App. D: a noise study (App. D.5) shows smooth degradation under Gaussian noise and clear gains from a noise-consistency loss, and we added analyses of latent-space interpolation, long-horizon rollouts, kernel sensitivity, and vertical sharing. These demonstrate stable behavior across spatial, temporal, and architectural variations.
> 2. **End-to-end optimization:** Please see our response to Question 1.
> 3. **Ablation studies:** Please see our response to Question 2.
> 4. **Equivariance of our model:** Full $E(3)$-equivariance is not imposed intentionally. As shown in App. A.2, Rayleigh–Bénard flow is horizontally but not vertically symmetric due to boundary conditions. Our architecture matches the correct group $(\mathbb{Z}^2,+)\rtimes D_4$, and enforcing larger groups would over-constrain the model.
> 5. **Baseline comparison:** We value fair comparison and therefore ensured that all models were trained under matched conditions, including identical $D_4$ augmentations. Thus, the baselines receive the same symmetry information through the data, while our model additionally incorporates it architecturally. This isolates exactly the question we aim to study: whether built-in equivariance provides benefits beyond augmentation alone. Additionally, building fully horizontally $D_4$-equivariant 3D variants of FNO or U-Net would require substantial additional methodological work and is orthogonal future research. Forecasting horizons differ only due to computational limits: FNO/U-Net cannot train beyond 5 steps at full resolution, while our latent predictor scales to 50 steps (Table 1). A 5-step $D_4$ model is also reported and performs competitively, supporting fairness.
> 6. **Long-horizon stability:** Beyond the 500-step rollouts in the main paper, App. D.4 now includes forecasts up to ~1400 steps. The $D_4$ model shows smooth, near-linear error growth that saturates rather than diverging, indicating stable long-term behavior.
>
> ### Questions
> 1. We tested several end-to-end variants (reconstruction, latent forecasting, consistency losses), all of which degraded long-term accuracy, indicating difficult loss balancing. Importantly, we did not observe negative effects from the two-stage procedure: Fig. 5–6 and App. D.4 shows smooth error growth up to 1400 steps. Partial finetuning remains a promising extension.
> 2. The main components ($D_4$-steerable convolutions, height-dependent kernels, vertical sharing) are tightly coupled, so isolating them requires re-engineering of full architectures, which is beyond the scope of this study. We did, however, perform an ablation on vertical parameter sharing. At our current domain height, removing it does not significantly change performance, while in larger vertical domains, it enables the model to scale while preserving approximate local symmetry.
> 3. All baselines were implemented in full 3D using standard reference code (e.g., neuralop) and trained with identical data, supervision, and optimizer settings; parameter counts were matched.
> 4. Thank you for this insightful question. Two new analyses were added: latent-space interpolation experiments (App. D.3) showing physically meaningful decoded trajectories, with errors that grow smoothly with temporal separation, and latent-pattern visualizations (App. D.2) revealing localized, oriented structures forming full $D_4$ orbits. These indicate a structured, interpretable latent space.
> 5. Our current model is trained per configuration (grid, aspect ratio, BCs), which is standard for CNN-based surrogates and not expected to be invariant to arbitrary changes. Broader generalization (e.g., varying $Ra$ or aspect ratio) can be achieved via parametric conditioning of kernels (Sec. 6.1). We already outlined this idea in Sec. 6.1 and view such parametric surrogates as a natural extension and promising avenue for future work.
> 6. App. D.5 now provides a noise-robustness analysis noise study: errors grow smoothly under Gaussian noise, and a noise-consistency loss significantly improves robustness for noise up to 0.05 and beyond. Although preliminary, these results show noise handling integrates naturally into our framework and is expected to improve with additional tuning.

---

### Author Response · Authors · 2025-11-24

We thank all reviewers for their constructive and insightful feedback. We are grateful for highlighting the value of our group-theoretic inductive biases, the structured and interpretable model pipeline, and the clarity of our quantitative evaluation (V5wy); for the recognition of our method’s high compression ratio, the elegance of our vertical parameter sharing strategy, and the mathematical rigor of our equivariance treatment (gjPD); and for commending the novelty of our architecture and the underlying technical contributions (Uzfy). This feedback has substantially improved the clarity, completeness, and empirical depth of the paper.

As reflected in the revised PDF, the key changes made in response to the reviewers’ comments are:
1. We have documented the full experimental setup for applying our architecture to a second PDE system, the Rayleigh–Taylor instability at $128\times128\times128$ resolution (Appendix D.1). This dataset is substantially larger and more turbulent than RBC. The experiments are still running due to the computational cost, but preliminary results are promising and indicate that the architecture transfers directly **without any modifications**, supporting its applicability to other systems with the same symmetry structure.
2. We added an extended long-horizon stability study (Appendix D.4). Autoregressive rollouts up to **1400 steps** show smooth, non-catastrophic error growth. The median RMSE increases approximately linearly and then **saturates**, indicating stable long-term behavior of the $D_4$-equivariant model far beyond the training window.
3. We included a comprehensive noise-robustness analysis (Appendix D.5). Evaluating the autoencoder under additive Gaussian noise shows graceful degradation, and introducing a **noise-consistency loss significantly improves robustness**: for noise levels up to 0.05, RMSE remains close to the clean baseline, and even at 0.1–0.2 the regularized model performs markedly better than the non-regularized one.
4. We expanded our latent representation analysis with new **latent-space interpolation** experiments and **latent pattern visualizations** (Appendices D.2–D.3). Interpolations decode to physically meaningful states with smoothly increasing error, and individual latent channels display localized, oriented structures forming full $D_4$ orbits, demonstrating that the model organizes features in a symmetry-consistent and interpretable way.
5. We conducted systematic sensitivity experiments on **height-dependent kernels** (Appendix D.6). Varying vertical kernel size and vertical resolution shows that reconstruction accuracy remains within a narrow range across all tested configurations, indicating that the height-dependent design is **robust to architectural choices**.
6. We added a detailed study of **vertical parameter sharing**, including early results at higher vertical resolution (Appendix D.7). All sharing configurations perform stably, with moderate and larger sharing giving slightly better accuracy at $N_3=32$. At higher resolution ($N_3=48$), sharing becomes **increasingly beneficial**, supporting its role in enabling scalable 3D equivariant models.
7. We have prepared an anonymized repository containing the full implementation of our architecture, including the equivariant CAE, convLSTM, height-dependent kernels, training and evaluation scripts, data generation and preprocessing code, reproduction instructions, and the hyperparameter configurations used in our experiments: https://anonymous.4open.science/r/surrogate-modeling-of-3D-rayleigh-benard-convection-with-equivariant-autoencoders

---

### Meta-Review · Area_Chair_BAQ8 · 2026-01-06

**Summary:**

The paper received generally positive but cautious evaluations, with reviewers acknowledging the technical soundness of the equivariant architecture and the thorough clarification provided during the rebuttal. The authors addressed several concerns related to equivariance design, baseline fairness, long horizon prediction stability, and aspects of latent space interpretation. However, a number of critical issues remain unresolved, most notably the reliance on a single physical system for evaluation, the absence of component-wise ablation studies, and limited evidence for broader generalization beyond the specific Rayleigh--Benard convection setup.

Given these remaining concerns and the lack of strong signals that reviewer scores would increase further, the expected final average rating is approximately 5.33, which is slightly lower than the borderline (5.5). At the same time, since two of the three reviewers provided positive ratings (6, 6, and 4 overall), I would not strongly object if the paper were accepted, leading to my choice "This decision can be bumped up (e.g., I am recommending a rejection but I wouldn't mind if the paper gets accepted.)".

**Reviewer Concerns:**

Reviewer V5wy

- Concerns that are potentially addressed:

    - End to end optimization: The authors clarified why a two stage training procedure was chosen and provided empirical evidence that long horizon stability is not negatively affected.

     - Equivariance of the proposed model: The authors explained why equivariance is enforced only in the horizontal plane, which is consistent with the underlying physics.

     - Fairness of baseline comparisons: Additional details were provided on baseline training and data usage.

     - Long horizon prediction stability: Extended autoregressive rollouts were added, demonstrating stable error growth.

     - Latent space interpretation: Additional latent space analyses were provided, partially addressing this concern.

- Concerns that might not be addressed:

    - Limited benchmark scope: All experiments are conducted on a single physical system, and the lack of real world or additional PDE benchmarks limits the strength of the conclusions.

    - Component-wise ablation: The contributions of individual architectural components are not fully isolated through systematic ablation.

    - Broader generalization: While broader generalization to new grids, boundary conditions, or physical parameters is discussed in the limitations, it is not empirically demonstrated.

Reviewer Uzfy

- Concerns that are potentially addressed:

    - Types of PDEs that could benefit: The authors provided discussion on applicability to other systems with similar symmetry structure.

    - Clarification of experimental setup: The experimental design was clarified, though this is closely related to the broader generalization concern raised by Reviewer V5wy.

- Concerns that might not be addressed:

    - Limited benchmark scope: Evaluation remains restricted to a single physical system of moderate scale.

Reviewer gjPD

- Concerns that are potentially addressed:

    - Code availability: The authors provided access to code and reproducibility resources.

    - Experimental setup clarification: Details on inclusion of solution snapshots and data augmentation for baseline training were clarified.

    - Long term forecasting across multiple Rayleigh numbers: Additional experiments were provided.

    - Sensitivity of height dependent kernels: New sensitivity analyses were included.

- Concerns that might not be addressed:

    - Limited benchmark scope: The evaluation remains focused on a single physical system without real world data.

     - Limited contribution breadth: While the height dependent kernel design is a meaningful technical contribution, other aspects of the method are viewed as more incremental and do not clearly constitute major technical advances on their own.


Common Concerns Across Reviewers

- The empirical evaluation is limited to a single physical system, which restricts confidence in the general applicability of the proposed method.

**Reviewer Scores:**

Reviewer V5wy: 6 with confidence 3

Reviewer Uzfy: 6 with confidence 3

Reviewer gjPD: 4 with confidence 4

While the authors provided clear and thorough responses to reviewer questions, several critical concerns remain, particularly regarding benchmark diversity, component wise ablation, and broader generalization. Given the current state of the discussion, it is unlikely that reviewer ratings would be updated further. This results in an expected final average rating of approximately 5.33.

---

### Decision · Program_Chairs · 2026-01-26

Reject